# High endogenous CCL2 expression promotes the aggressive phenotype of human inflammatory breast cancer

Anita Rogic [1,3], Ila Pant [2,3,7], Luca Grumolato [2,3,4,7], Ruben Fernandez-Rodriguez [1,3], Andrew Edwards[1,3], Suvendu Das[1,3,6], Aaron Sun [1,3], Shen Yao[2,3], Rui Qiao[2,3], Shabnam Jaffer [2], Ravi Sachidanandam[2,3], Guray Akturk [3], Rosa Karlic [5], Mihaela Skobe [1,3,8✉] & Stuart A. Aaronson [2,3,8✉]

Inflammatory Breast Cancer (IBC) is a highly aggressive malignancy with distinct clinical and histopathological features whose molecular basis is unresolved. Here we describe a human IBC cell line, A3250, that recapitulates key IBC features in a mouse xenograft model, including skin erythema, diffuse tumor growth, dermal lymphatic invasion, and extensive metastases. A3250 cells express very high levels of the CCL2 chemokine and induce tumors enriched in macrophages. CCL2 knockdown leads to a striking reduction in macrophage densities, tumor proliferation, skin erythema, and metastasis. These results establish IBC-derived CCL2 as a key factor driving macrophage expansion, and indirectly tumor growth, with transcriptomic analysis demonstrating the activation of multiple inflammatory pathways. Finally, primary human IBCs exhibit macrophage infiltration and an enriched macrophage RNA signature. Thus, this human IBC model provides insight into the distinctive biology of IBC, and highlights potential therapeutic approaches to this deadly disease.

[1] Laboratory of Cancer Lymphangiogenesis, Department of Oncological Sciences, Icahn School of Medicine at Mount Sinai, New York, New York, USA. [2] Department of Oncological Sciences, Icahn School of Medicine at Mount Sinai, New York, New York, USA. [3] Tisch Cancer Institute, Icahn School of Medicine at Mount Sinai, New York, New York, USA. [4] Normandie University, UNIROUEN, INSERM, DC2N Rouen, France. [5] Bioinformatics group, Division of Molecular Biology, Department of Biology, Faculty of Science, University of Zagreb, Zagreb, Croatia. [6] Present address: Institute of Advanced Research, Department of Biological Sciences and Biotechnology, Koba Institutional, Area, Gandhinagar 382 426, Gujarat, India. [7] These authors contributed equally: Ila Pant, Luca Grumolato. [8] These authors jointly supervised the work: Mihaela Skobe, Stuart A. Aaronson. ✉email: mihaela.skobe@mssm.edu; stuart.aaronson@mssm.edu

nflammatory Breast Cancer (IBC) is the most aggressive form of breast cancer, with the lowest survival rate compared to all other breast cancers. At diagnosis, these tumors are considered at least stage IIIB, with most patients exhibiting extensive lymph node metastases and nearly 30% distant metastases[1–5]. Clinical symptoms include skin redness, an orange-peel appearance (*peau d'orange*) and swelling, making it initially challenging to distinguish from non-cancerous inflammatory diseases of the breast, as these tumors also grow diffusely rather than as palpable nodules. Another prominent feature of IBC is the presence of tumor emboli in dermal lymphatic vessels[4]. Diagnosis of IBC is made based on biopsy confirming the presence of cancer cells either in dermal lymphatics or in the breast parenchyma associated with clinical detection of inflammation[5,6].

These distinct clinical and histopathological features suggest unique molecular drivers of the IBC phenotype. However, despite many efforts over the past decade to find unique molecular characteristics, IBC has not been defined as a separate molecular subtype. Genomic studies have revealed that IBC is a heterogeneous disease comprising all classical breast cancer molecular subtypes[7,8]. Even within triple-negative tumors, the most common IBC subtype, the seven triple-negative molecular phenotypes identified exist in similar proportions in IBC and non-IBC, with no robust differences in the gene expression signature between IBC and non-IBC[4,7,8].

One of the key barriers in advancing understanding of IBC has been the lack of models which recapitulate all of the unique features of human IBC. To date, there are only a few human IBC-derived cell lines[9–14]. The paucity of models that reproduce features of human IBC upon xenotransplantation and are amenable to manipulation has been a major impediment to the mechanistic understanding of unique aspects of the IBC phenotype. Here, we describe a triple-negative human IBC cell line that recapitulates all key features of human IBC in a mouse xenotransplant model. We show that IBC tumors mobilize distinct subsets of inflammatory cells locally and systemically, and demonstrate a key role of CCL2 secreted at very high levels in recruiting macrophages into tumors and thereby facilitating tumor growth and metastasis.

## Results

### Biological characterization of the A3250 human IBC cell line.
The A3250 breast carcinoma cell line was established from a primary tumor of a patient with triple negative IBC, as previously described for other A series lines[15]. Short Tandem Repeat (STR) analysis confirmed that this line was unique based on its comparison to all human cell lines in ATCC STR and DSMZ STR databases, with less than 55% similarity to any of the existing STR profiles (Supplementary Table 1). In tissue culture, confluent A3250 cells exhibited epithelial morphology (Fig. 1a). At lower cell densities, we observed two distinct phenotypes: cell colonies consisting of epithelial-like polygonal cells in close contact, and colonies comprised of scattered cells with more elongated appearance (Fig. 1b). A3250 cells exhibited higher colony-forming ability than MDA-MB-231 cells, a commonly studied triple negative breast cancer (TNBC) line and retained the ability to form colonies even at low serum concentration (1% FBS) and at very low seeding densities (Supplementary Fig. 1a, b). In monolayer culture, A3250 cells exhibited a population-doubling time of 28 h in 10% FBS and proliferation was not significantly reduced upon short-term culture in 1% serum (Supplementary Fig. 1c). They lacked detectable estrogen or progesterone receptors (ER or PR), and expressed *HER2 (*Human Epidermal Growth Factor Receptor-2, *ERBB2)* mRNA at levels comparable to other TNBC cell lines (Supplementary Fig. 1d). In comparison, BT474 cells with amplified HER2 expressed more than 70-fold higher levels (Supplementary Fig. 1d, Supplementary Table 2). A3250

orthotopic tumors also lacked ER, PR or HER2 protein, in contrast to MCF7 or BT474 tumors that stained positive for ER and PR or HER2, respectively (Supplementary Fig. 1e–j). These data confirmed that A3250 was a TNBC cell line.

### A3250 human xenograft model recapitulates key clinical and histopathological features of human IBC.
We engineered A3250 cells to express green fluorescent protein (GFP) and firefly luciferase (Fluc) to allow for analyses at single-cell resolution in tissue sections, and for quantitative analyses of tumor burden by bioluminescence imaging, respectively. Orthotopic injection of $10^6$ cells into the mammary fat pad of either SCID or NOD SCID mice led to the formation of primary tumors at 100% incidence, with comparable growth kinetics (Fig. 1c, d). A3250 tumors induced pronounced erythema in the overlying skin (Fig. 1e). Nipple changes observed in IBC patients, such as inversion and protrusion, were also seen early in A3250 tumor development (Supplementary Fig. 2a–c). Tumors did not form typical palpable nodules characteristic of non-IBCs, but were flat in appearance (Fig. 1e, f). Analysis of GFP-tagged A3250 tumors revealed a diffuse distribution of tumor cells as clusters or single cells, throughout the mammary gland and adjacent skin (Fig. 1g, h, Supplementary Fig. 2d, e). Accordingly, A3250 tumors had abundant stroma and low tumor-stroma ratio (% tumor cell area: 33% ± 10.6; range 18–43%, $n = 3$). By comparison, both SUM149 IBC and MDA-MB-231 non-IBC tumors formed nodules (Supplementary Fig. 2f–j) and exhibited high tumor-stroma ratio (% tumor cell area: SUM149, 61% ± 3.44, range 55–64%, $n = 4$; MDA-MB-231, 77% ± 2.0; range 74–79%, $n = 4$). SUM149 formed many invasive clusters (Supplementary Fig. 2i), but did not show the dispersed growth pattern observed with A3250. Because A3250 tumors exhibited a diffuse pattern of growth, bioluminescence imaging provided more accurate assessment of tumor burden than volume measurements.

Histopathological evaluation of hematoxylin and eosin (H&E) stained tumor sections indicated poorly differentiated carcinoma with pleiotropic atypical nuclei (Fig. 1i), showing extensive invasion into the subcutaneous fat (Fig. 1j), indicating its aggressive nature. A3250 tumors exhibited prominent dermal lymphatic vessel invasion (LVI), one of the key characteristics of IBC. Lympho-vascular emboli were frequently seen in greatly enlarged dermal lymphatic vessels (Fig. 1k). In contrast, we did not observe dermal LVI in SUM149 tumors at four weeks (Supplementary Fig. 2j), despite their highly invasive nature. Immuno-histochemical (IHC) staining revealed keratin expression throughout the A3250 tumors, with higher intensity signal in tumor emboli in lymphatic vessels (Fig. 1l). A3250 tumors also showed high mitotic rates based on Ki67 staining (Fig. 1m), defined as ≥10% of Ki67+ tumor cells[16]. Analysis of tumor cell proliferation by EdU (5-ethynyl-2′-deoxyuridine) pulse independently revealed a high proliferation rate, with an average of 458 Edu+ tumor cells per mm² tumor/6 hr (STD+/−70.2) (Supplementary Fig. 3a–f, Supplementary Table 3).

Ingenuity Pathway Analysis (IPA) of mRNA Sequencing (mRNA-Seq) data revealed that pathways implicated in cell growth and protein synthesis, in particular mTOR signaling and its effectors, and translation initiation factors eIF2, eIF4 and p70S6K signaling were among those most highly activated in A3250 cells. Among the top ten were also pathways related to cell metabolism, such as mitochondrial dysfunction, oxidative phosphorylation and sirtuin signaling, as well as integrin signaling and junction remodeling (Fig. 1n and Supplementary Data 1). The majority of these pathways were conserved in A3250 cells in vitro and tumors in vivo.

### A3250 IBC tumors spontaneously metastasize to lymph nodes and distant organs.
Rapid and extensive regional and distant

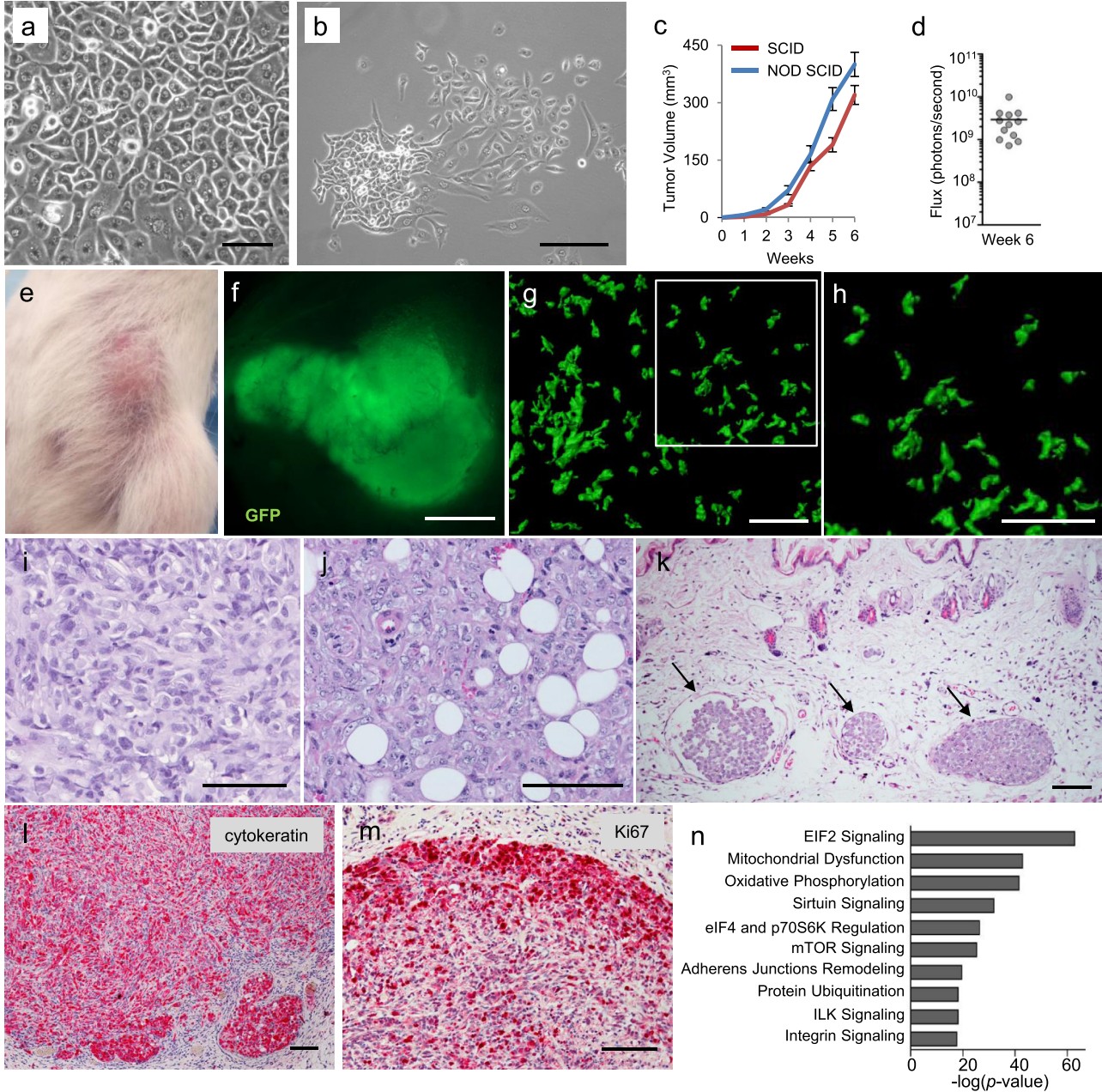

**Fig. 1 Characteristics of A3250 tumor cells in vitro and tumors in vivo. a** Bright-field microscopy of confluent A3250 tumor cells in vitro. **b** Morphology of A3250 tumor cell clones in vitro. Images are representative of three independent experiments. **c** Tumor growth following orthotopic injection of $10^6$ A3250 tumor cells in SCID or NOD SCID mice measured by caliper weekly. NOD SCID $n = 8$ and SCID $n = 6$ biologically independent mice. Data represent mean ± SEM (Standard Error of the Mean). **d** Bioluminescence imaging of A3250 tumors in SCID mice at week six ($n = 12$ mice). Black line denotes mean value. **e** Representative picture of an A3250 tumor at 6 weeks showing erythema of the skin and flat tumor appearance. **f** Fluorescence stereo-microscopy of an A3250 tumor (GFP, green) ex vivo. **g** A3250/GFP 300 μm tumor section imaged by confocal microscopy and reconstructed in 3D. Note diffuse pattern of growth. (GFP, green) (**h**). Higher magnification of boxed area in (**g**). H&E staining of paraffin embedded tumor sections (**i**, **j**) and overlying skin (**k**). Note the presence of lymphovascular emboli in the dermis (**k**, arrows). IHC staining for cytokeratin (**l**) and for the proliferation marker Ki-67 (**m**). Images are representative of three independent experiments. $n = 4$ biologically independent mice. **n** Top ten A3250 canonical pathways based on mRNA-Seq of A3250 tumors, determined by IPA. Top pathways are ranked based on $p$ values. Scale bars: **a**, **b**, 100 μm; **f**, 2 mm; **g**–**m**, 100 μm. Source data are provided as a Source Data file.

metastasis is a salient feature of human IBC. In accordance with the notion that A3250 tumors retained key characteristics of human IBC, A3250 tumor xenografts spontaneously metastasized to multiple organs upon orthotopic injection into the mammary fat pad. This was determined by bioluminescence analysis of organs ex vivo and validated by analysis of GFP signal by stereomicroscopy of whole organs and tissue sections (Fig. 2). A3250 cells metastasized to the sentinel (inguinal) lymph nodes (LNs) at 100% incidence and to more distant LNs (axillary and brachial) at 46 % incidence within six weeks (16/35 LNs positive) (Fig. 2a). Lymph node metastases commonly presented as large, highly proliferative lesions, and in the majority of the cases almost completely replaced the lymph node (Fig. 2b, d–f).

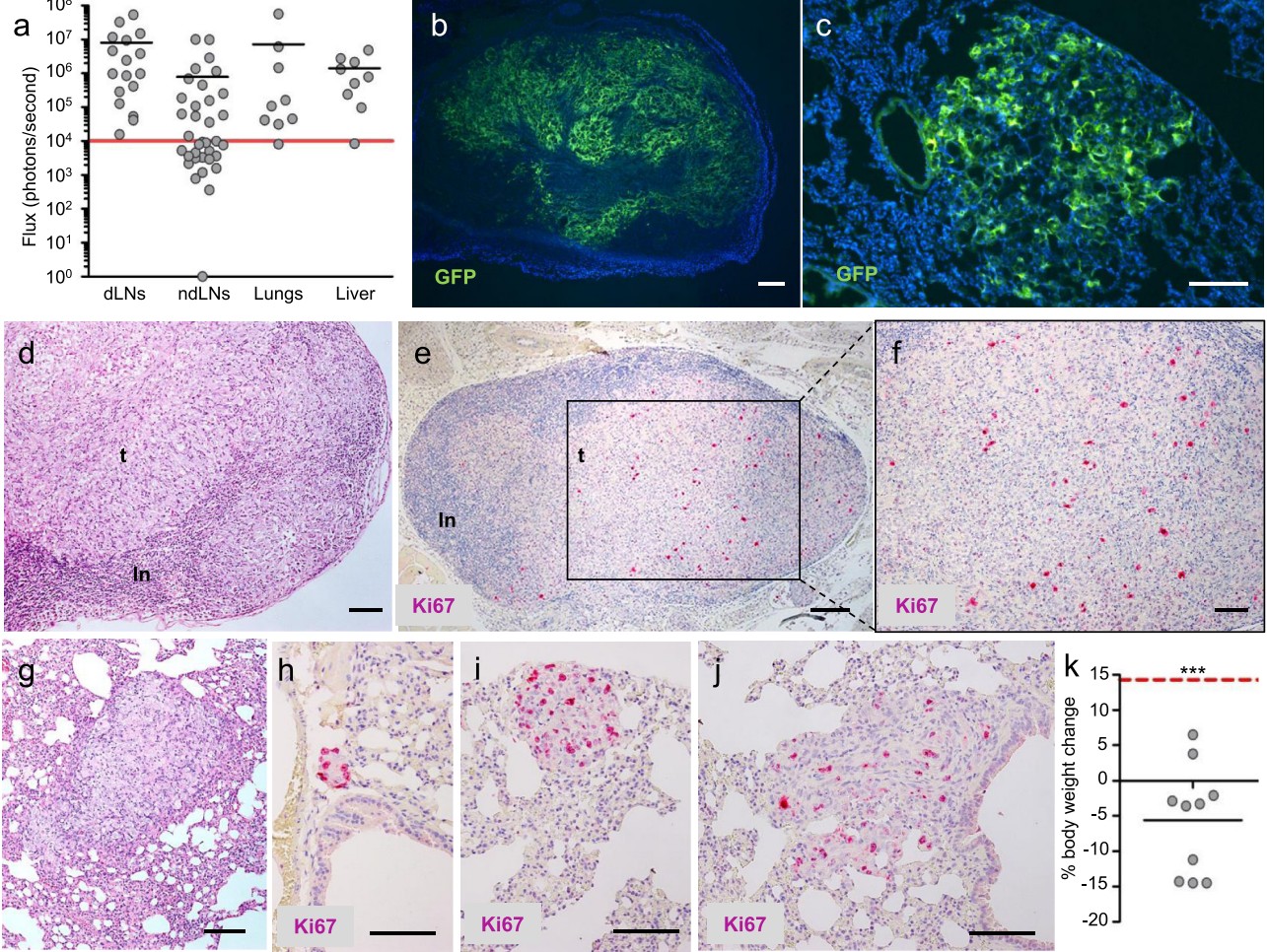

**Fig. 2 Features of metastases in A3250 xenograft model. a** Metastatic burden in organs as indicated, determined by bioluminescent imaging (total Flux) ex vivo, 6 weeks post-injection ($n = 9$ mice). Inguinal draining lymph nodes (dLNs; $n = 17$), axillary and brachial non-draining lymph nodes (ndLNs; $n = 35$). Solid black lines indicate mean values. Red solid line indicates the threshold of positive Fluc signal above background. Representative images of lymph node (**b**) and lung (**c**) metastases (GFP, green) in tissue sections (DAPI, blue). Metastases in the lymph node stained by H&E (**d**) or Ki-67 proliferation marker (**e**, **f**). Lung metastases stained by H&E (**g**) or Ki-67 (**h–j**). Different sizes of metastases are depicted (**h–j**). **k** Weight loss of mice bearing A3250 tumors at six-week endpoint ($n = 10$ biologically independent mice). Differences between the true means were determined through a one-sample $t$ test ($p < 0.0001$). Short black line indicates mean % weight loss. Red dotted line indicates the average weight gain of an age-matched, naïve control mouse during the same period. $p$ value was determined by one sample $t$ test. Images are representative of two independent experiments. $n = 5$ biologically independent mice. ***$p < 0.001$. Scale bars: **b**, **e**, 200 μm; **c**, **d**, **f–j**, 100 μm. Source data are provided as a Source Data file.

A3250 tumors also showed a high incidence of metastases to the lung and liver (89%) (Fig. 2a). Lung lesions exhibited a unique growth pattern, with cell clusters distributed diffusely through the lung parenchyma (Fig. 2c), closely resembling the diffuse pattern of primary tumor growth. Lung metastases were large (Fig. 2g), frequently replacing an entire part of the lobe, and were highly proliferative (Fig. 2h–j). Proliferation was particularly high in small lesions (Fig. 2h–j). We did not observe solitary disseminated cells or micro-metastases at six weeks post-injection, indicating that upon seeding, A3250 cells proliferated rapidly and colonized the lung. Associated with the high metastatic burden at endpoint, we observed notable weight-loss (Fig. 2k) and muscle-wasting of tumor-bearing animals, consistent with cachexia. Together, these findings indicate that A3250-derived tumors recapitulated the aggressive metastatic phenotype of human IBC.

**Transcriptomic profiling reveals activation of multiple pathways associated with inflammation in A3250 cells.** To gain insight into potential drivers of its aggressive IBC phenotype, we performed transcriptomic analyses of A3250 cells. We compared the A3250 gene expression profile with that of six human TNBC cell lines; two IBC and four non-IBC. Hierarchical clustering of mRNA-Seq data revealed that the transcriptional profile of A3250 cells was closest to that of SUM149 IBC cells (Fig. 3a). There were 4881 Differentially Expressed Genes (DEGs) in A3250 when compared to MDA-MB-231, 7033 DEGs compared to IBC3, and 4919 DEGs compared to SUM149. There were 1779 DEGs in A3250 when compared to all three cell lines, MDA-MB-231, SUM149 and IBC3 (Fig. 3b).

IPA revealed that in comparison to MDA-MB-231 cells, A3250 displayed remarkable enrichment in pathways associated with inflammation. There were ten canonical pathways activated in A3250 compared to MDA-MB-231, of which eight were associated with inflammation (Fig. 3c, Supplementary Data 2). The wound healing pathway was most significantly upregulated, followed by the tumor microenvironment (-log($p$ value): 9.99 and 8.12, respectively), whereas TREM1 signaling showed the highest $z$-score (3.58). The TREM1 pathway triggers and amplifies inflammation through interactions with TLR pathways, and is

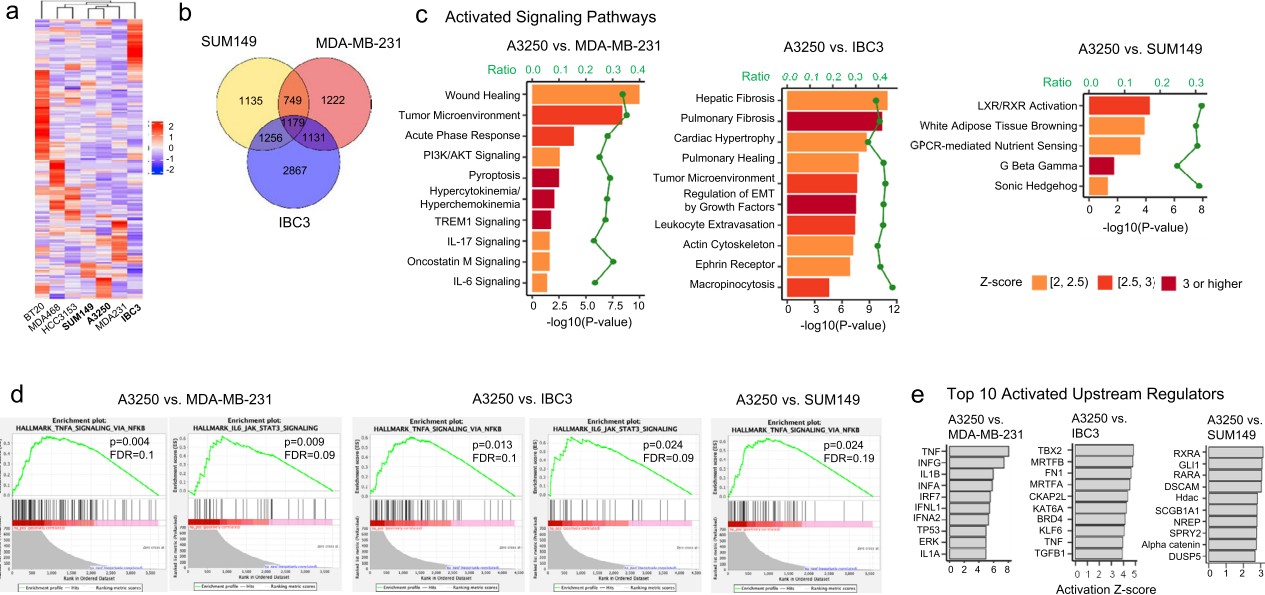

**Fig. 3 Transcriptional profile of A3250 compared to MDA-MB-231, IBC3 and SUM149 cells. a** Hierarchical clustering of 175 top differentially expressed genes (DEGs) by mRNA-Seq between A3250, IBC3, and SUM149 IBC cells and several non-IBC cell lines. IBC cell lines are in bold. Z-score is indicated. **b** Venn diagram of DEGs in A3250 compared to MDA-MB-231, SUM149, and IBC3. **c** Pathways activated in A3250 in comparison to MDA-MB-231, IBC3, and SUM149 by IPA. **d** Selected Gene Set Enrichment Analyses (GSEA) for gene sets enriched in A3250 compared to MDA-MB-231, IBC3, and SUM149. **e** Upstream regulators differentially activated in A3250 compared to MDA-MB-231, IBC3, and SUM149, determined by IPA.

upstream from NF-κB. Other top enriched canonical pathways included hypercytokinemia (cytokine storm), IL-6, oncostatin M (member of the IL-6 family), IL-17, acute phase response signaling and pyroptosis, which are all related to inflammation and perturbation of the immune system. Among pathways inactivated in A3250 compared to MDA-MB-231, several were related to neuronal signaling, such as semaphorin, netrin and CREB signaling in neurons (Supplementary Data 2).

In comparison to IBC3, A3250 showed activation of pathways related to fibrosis, EMT and cytoskeleton among others (Fig. 3c, Supplementary Data 3). Activation of PI3/AKT signaling and tumor microenvironment, and inactivation of semaphorin signaling pathway were observed in A3250 in comparison to both, MDA-MB-231 and IBC3. A3250 showed only five pathways enriched in comparison to SUM149, of which LXR/RXR canonical pathway, related to lipid metabolism and inflammation was most significantly upregulated (Fig. 3c, Supplementary Data 4).

We next performed Gene Set Enrichment Analysis (GSEA) by using the hallmark collection of molecular signature databases (MSigDB), which capture and refine information from multiple gene sets and show high degrees of association with the corresponding protein profiles[17]. GSEA showed that from 50 hallmark datasets, 34 gene sets were significantly enriched in A3250 compared to MDA-MB-231, 31 gene sets compared to IBC3, and 22 gene sets compared to SUM149. Among the pathways implicated in inflammation, hallmark "TNFA signaling via NFKB" was significantly enriched in A3250 compared to MBA-MB-231, IBC3 and SUM149 (Fig. 3d). Hallmark "IL6 JAK STAT3 signaling" was significantly enriched in A3250 compared to MDA-MB-231 and IBC3 (Fig. 3d) and only marginally compared to SUM149 ($p = 0.098$, FDR = 0.12), indicating that JAK/STAT3 pathway is activated in A3250 and in agreement with previous studies implicating the JAK/STAT3 pathway in IBC[5].

Consistent with our evidence indicating activation of multiple inflammatory pathways in A3250 by IPA and GSEA, analysis of activated upstream regulators showed a striking association with

inflammation in comparison to MDA-MB-231 (Fig. 3e, Supplementary Data 5). The top differentially activated upstream regulator was TNF (z- score 8.0; p value of overlap 1.86E–50), followed by other upstream regulators driving inflammation and regulating innate immunity. Prominent categories included interferons (INFG, INFA, INFA2, INFL1), interferon regulatory factors (IRF7, IRF3, STING), interleukins (IL1A, IL1B) and TLRs (TLR4, TLR7) (Fig. 3e, Supplementary Data 5). IRF7 for example is a key transcriptional regulator of type I INF-dependent immune responses and plays a critical role in innate immune responses. Among the top inactivated upstream regulators was RC3H1 that modulates activity of the NF-κB pathway and restrains inflammation by suppressing TNF expression[18]. TNF was also among the top ten upstream regulators activated in A3250 compared to IBC3 (Fig. 3e, Supplementary Data 6), but not to SUM149 (Fig. 3e, Supplementary Data 7). Taken together, these data demonstrate prominent activation of pathways associated with inflammation and innate immunity in A3250 IBC compared to MDA-MB-231 non-IBC cells, whereas differences in inflammatory pathways between A3250 and IBC3 or SUM149 IBC were less pronounced.

**A3250 IBC tumors induce prominent regional and systemic inflammation.** Pathologists have divergent views about the extent and significance of inflammatory infiltrate in IBC[19], and the prevailing view is that inflammation is a consequence of the blockage of dermal lymphatic vessels by cancer cells. Thus, we sought to characterize the kinetics of inflammatory cell accumulation in A3250 tumors in comparison to LVI. Immunostaining demonstrated that A3250 tumors were heavily infiltrated with CD45+ cells four weeks post-injection, in comparison to MDA-MB-231 tumors (Fig. 4a, b). In the dermis, CD45+ cells were already abundantly present at three weeks post-injection (Fig. 4c). Dermal lymphatics were numerous and enlarged at that same time-point, but did not yet contain tumor emboli (Fig. 4d). A high

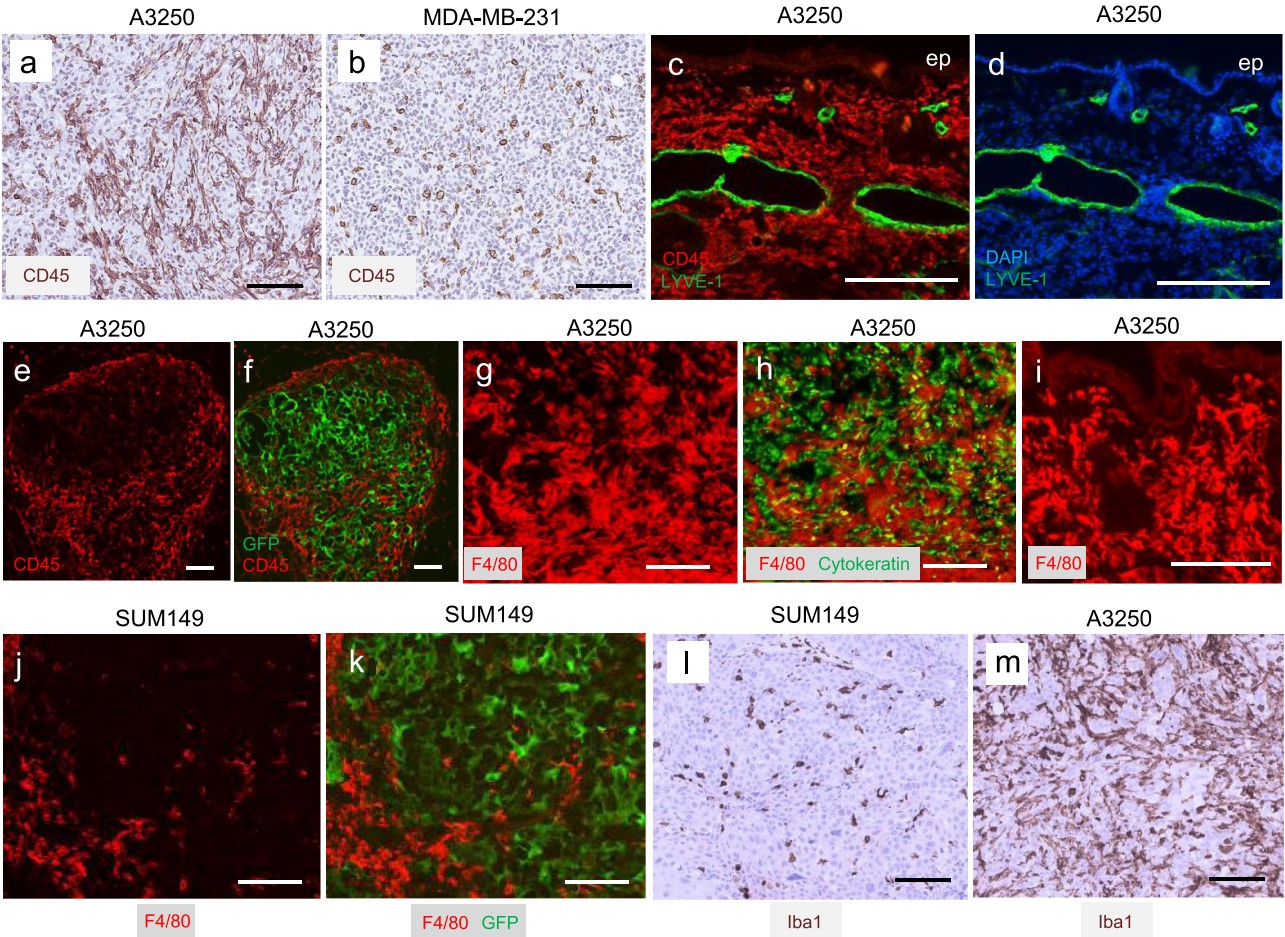

**Fig. 4 Inflammation associated with A3250 tumors by immunostaining analysis. a, b** CD45 IHC staining of tumors at four-weeks as indicated. **c, d** Immunofluorescent staining of the skin adjacent to A3250 tumors for CD45 (red) and lymphatic marker LYVE-1 (green) at three-weeks (DAPI, blue). Note dense inflammatory infiltrate in the dermis and the absence of tumor cells within the lymphatics at this time point. **e, f** Immunofluorescent staining of A3250 tumor (GFP, green) for CD45 (red) at one-week post-injection. **g, h** Immunofluorescent staining of A3250 tumor for F4/80 macrophage marker (red) and cytokeratin (green) at four weeks as indicated. **i** Skin adjacent to A3250 tumor immunostained for F4/80. **j, k** Immunostainining of SUM149 tumor (GFP, green) for F4/80 (red) at four-weeks. **l, m** Iba1 IHC staining of tumors at four-weeks as indicated. Images are representative of three independent experiments. $n = 3$ biologically independent mice. Scale bars: 100 μm.

density of CD45$^+$ cells was also observed in the dermis of the skin overlying A3250 tumors, even when the tumor was distant from the skin, separated by a layer of muscle and fat (Supplementary Fig. 4a–c). Striking accumulation of CD45$^+$ cells in A3250 tumors was seen as early as a week post-inoculation (Fig. 4e, f), concomitant with lymphatic vessel dilation in the dermis, but prior to detectable tumor invasion of lymphatic vessels. These results demonstrate that inflammatory cells accumulated prior to tumor invasion of dermal lymphatics. Thus, the abundance of inflammatory cells in A3250 xenografts likely resulted from their active recruitment by tumor-derived factors, rather than as a consequence of the blockage of lymphatic vessels by tumor emboli.

Immunostaining further revealed a very high density of F4/80$^+$ macrophages throughout the viable A3250 tumor and in the adjacent dermis (Fig. 4g–i, Supplementary Fig. 4d–f). In comparison, both SUM149 (Fig. 4j–m, Supplementary Fig. 4g–i) and MDA-MB-231 tumors (Supplementary Fig. 4j–l) had much lower macrophage content. Quantification of F4/80$^+$ macrophages in tumor sections revealed ~3-fold increase in A3250 tumors compared to either SUM149 or MDA-MB-231 (Supplementary Fig. 4m). Very few Ly6G$^+$ neutrophils were detected in viable areas of A3250 or SUM149 tumors, in contrast to MDA-MB-231 tumors

which were uniformly infiltrated (Supplementary Fig. 4n–p). Immunostaining with F4/80 also revealed high density of macrophages in the skin overlying A3250 tumors in comparison to MDA-MB-231 (Supplementary Fig. 4q–s).

Flow cytometry demonstrated that A3250 tumors contained two major populations of CD11b$^+$ myeloid cells at four weeks post-injection (Fig. 5a–c, Supplementary Fig. 5a, b), based on Ly6C and Ly6G in combination with other markers to define the myeloid lineage subsets[20–22]. Tumor-associated macrophages (TAMs, defined as CD45$^+$CD11b$^+$ F4/80$^+$Ly6C$^-$Ly6G$^-$) were most abundant (avg 32.2%, median 31.7%), followed by Ly6C$^{hi}$ inflammatory monocytes (avg 19.5%, median 19.6%) (CD45$^+$CD11b$^+$Ly6C$^{hi}$F4/80$^{lo/-}$Ly6G$^-$). In contrast, tumor-associated neutrophils (TANs, defined as CD45$^+$CD11b$^+$ Ly6G$^+$Ly6C$^{lo}$F4/80$^-$) were the least represented CD11b$^+$ subset (avg 17.6%, median 9.6%) (Fig. 5c). A3250 tumors had ~3-fold higher proportion of TAMs and ~3-fold lower proportion of TANs compared to MDA-MB-231 tumors. In contrast, TANs were the most highly represented subset in MDA-MB-231 tumors, and the proportion of TAMs was the lowest (Fig. 5d–f).

We next examined lungs from mice bearing A3250 tumors at four weeks post-injection to gain insights into the

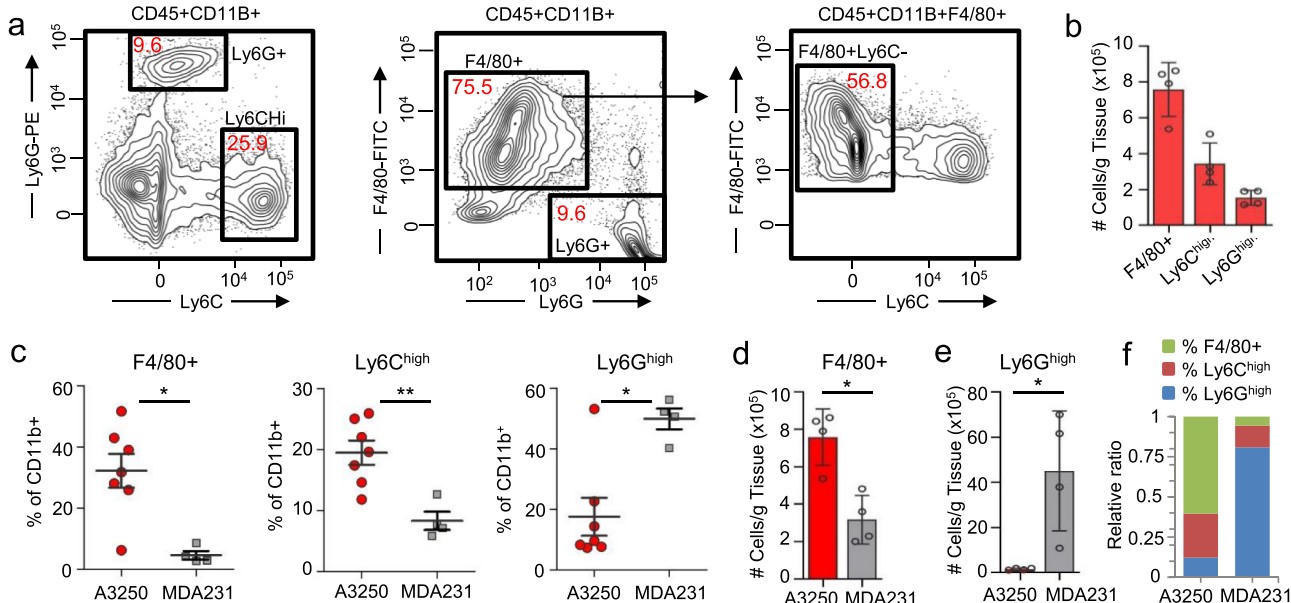

**Fig. 5 Inflammation associated with A3250 tumors by FACS analysis. a** Gating strategy used to characterize myeloid cell populations in A3250 tumors. Numbers in red indicate percent of total $CD45^+$ $CD11b^+$ cells. **b** Quantification of different myeloid cell subsets in A3250 tumors at four weeks ($n = 4$). **c** Comparison of the frequency of myeloid cell subsets in A3250 ($n = 7$) and MDA-MB-231 ($n = 4$) tumors at 4 weeks as indicated. To control for repeated measures, a Linear Mixed Model fit by REML and $t$ test using Satterthwaite's method was used for analysis (%F4/80+: $p = 0.015927$, %Ly6C$^{high}$: $p = 0.00625$, %Ly6G$^{high}$: $p = 0.0205$). Quantification of F4/80$^+$ macrophages (**d**) and neutrophils (**e**) in A3250 ($n = 4$) and MDA-MB-231 ($n = 4$) tumors. Two-tailed Mann–Whitney $t$ test yielded $p = 0.0286$ (**d**) and $p = 0.029$ (**e**). **f** Relative ratios of different myeloid cell populations in tumors as indicated. On the charts, mean ± SEM is indicated.**$p < 0.01$, *$p < 0.05$. Source data are provided as a Source Data file.

microenvironment of metastases. H&E staining revealed abnormal lung architecture characterized by areas of dense inflammatory infiltrate and loss of alveolar structures (Fig. 6a–d). Immunostaining showed that $CD45^+$ inflammatory cells were associated with disseminated A3250 tumor cells, and were more prominent than in the lungs of mice bearing MDA-MB-231 tumors of comparable size (Fig. 6e–h). Flow cytometry demonstrated a trend towards an increase of Ly6C$^{hi}$ myeloid cells and a ~40% increase in Ly6G$^{hi}$ neutrophils compared to the lungs of tumor-naïve mice (Fig. 6i). Fractions of total $CD11b^+$ cells, Ly6C$^{hi}$ inflammatory monocytes and Ly6G$^{hi}$ neutrophils were significantly increased in blood in comparison to tumor-free mice (Fig. 6j), and this correlated with the increase of these cell subsets in the lung. In comparison to MDA-MB-231, Ly6G$^+$ cells were significantly increased and there was a trend towards increase of Ly6C$^+$ cells in blood. The fraction of Ly6G$^+$ cells relative to F4/80$^+$ and Ly6C$^+$ cells was lower in lungs with A3250 metastases than in the lungs with MDA-MB-231 metastases (Fig. 6k), because the total number of Ly6G$^+$ cells was 2.8-fold lower (Avg. number of lung Ly6G$^+$ cells/g tissue: A3250, $3.7 \times 10^6$ vs. MDA-MB-231, $14.2 \times 10^6$). Of note, despite an increase of Ly6G$^+$ neutrophils in the blood and lungs, there were very few Ly6G$^+$ cells within A3250 primary tumors, suggesting a mechanism for selective exclusion of neutrophils. In contrast, in the MDA-MB-231 tumor model Ly6G$^+$ neutrophils were the main cell subset recruited to both, primary tumors (Fig. 5c, e, f) and to the lungs (Fig. 6k).

**CCL2 is highly expressed in A3250 cells and in human IBCs.** To identify factors that might mediate the florid recruitment of myeloid cells by A3250 tumors, we analyzed chemokine expression in a mRNA-Seq data set generated from A3250 tumors and cells in vitro. Analysis of human chemokines[23] expressed by the A3250 tumors revealed expression of 10 out of 44 chemokines

(FPKM ≥ 1 in each sample; Fragments Per Kilobase of exon per Million mapped reads), of which only three, CCL2, CXCL5, and CCL20, were expressed at high levels both in vivo and in vitro (FPKM > 10) (Fig. 7a, Supplementary Data 8). By comparison of their expression levels in tumors and cell culture in the same experiment, CCL2 was the highest expressed chemokine by mRNA-Seq (388 and 467 FPKM, respectively) and by quantitative Polymerase Chain Reaction (qPCR) (Fig. 7a, b; Supplementary Data 8). SUM149 cells expressed several chemokines from the CXCL family that were also expressed by A3250, including CXCL1, 2, 3 5, and 16, but did not express CCL2 (Fig. 7b). CCL2 mRNA was also highly upregulated in A3250 cells not only in comparison to SUM149, but also to IBC3 and to a panel of non-IBC breast cancer cell lines (Fig. 7c, Supplementary Table 4). High levels of CCL2, CXCL5 and CCL20 proteins were confirmed in supernatants of A3250 cells by Olink multiplex immunoassay. SUM149 IBC cells also secreted CXCL5 and CCL20, but not CCL2 protein, whereas MDA-MB-231 expressed either no detectable or very low levels of this chemokine (Fig. 7d, Supplementary Table 5). While these results correlated relatively well with qPCR or mRNA-Seq, any differences could reflect cell specific variations in RNA translational efficiency or protein secretion.

To investigate how the chemokine expression profile of A3250 cells related to that of human breast cancers, we analyzed expression levels of the top three most highly expressed chemokines, CCL2, CXCL5, and CCL20, in the TCGA-BRCA database of human invasive breast cancer samples of which none were annotated as IBC ($n = 1222$). All three chemokines were expressed at much higher levels in A3250 IBC compared to their median expression levels in TCGA breast cancer samples (Fig. 7e). To interrogate expression of CCL2 in IBC patient samples, we analyzed a Van Laere microarray data set consisting of 41 human IBCs and 55 non-IBCs[24,25], and an Iwamoto data set consisting of 25 IBCs and 57 non-IBCs[26]. These data sets were chosen because they had most IBC samples and because mRNA-Seq data for human IBCs have

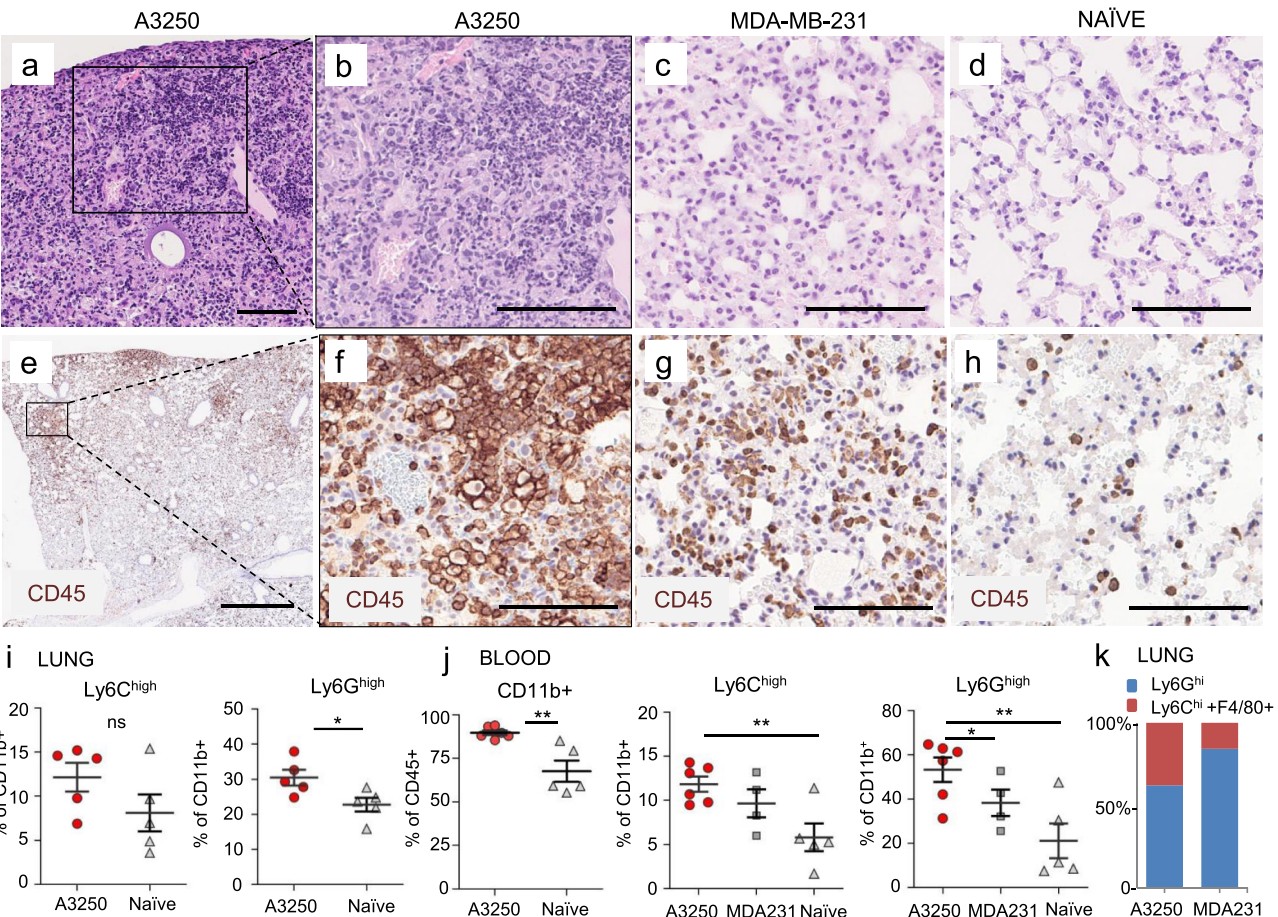

**Fig. 6 Lung inflammation in mice bearing A3250 tumors. a–h** Lungs of mice bearing A3250 or MDA-MB-231 tumors at 4 weeks post-injection, and tumor-naïve lungs from age-matched SCID mice as indicated. Sections were stained with H&E (**a–d**) or immuno-stained for CD45 (**e–h**). **i** Frequency of myeloid cell subsets in lungs from A3250 tumor-bearing and naïve mice determined by flow cytometry (%Ly6C: $p = 0.09$, %Ly6G: $p = 0.0146$). $n = 5$ lungs in each experimental group. **j** Frequency of myeloid cell subsets in blood from tumor-bearing and naïve mice as indicated, determined by flow cytometry (%CD11b+: $p = 0.00373$; %Ly6C+: $p = 0.00629$; %Ly6G+: $p = 0.00728$). $n = 5$ blood samples for naïve mice and 6 for mice with A3250 tumors A two-tailed $t$ test was performed to determine significance (**i, j**). **k** Ratio of myeloid cell subsets in lungs as indicated. Black line on the plots indicates mean ± SEM. Scale bars: (**a**), (**e**), 1 mm; (**b–d**), (**f–h**), 100 μm.**p < 0.01 *p < 0.05; ns, not significant. Source data are provided as a Source Data file.

not yet been reported. *CCL2* median expression levels were ranked significantly higher in IBCs than in non-IBCs in both datasets (Fig. 7f, g). Taken together, these data indicate that human IBCs, like A3250, commonly exhibit high CCL2 expression levels.

**Chemokine receptor CCR2 is highly expressed by myeloid cells in the A3250 IBC tumor model.** CCL2 exerts its effects through CCR2, which is the main chemokine receptor mediating monocyte and macrophage recruitment to sites of inflammation[27,28]. We investigated the expression of CCR2 in A3250 tumors, lungs and on leukocytes in the blood. In the blood, CCR2 was expressed by ~30% of total CD11b+ cells, and mainly by Ly6Chi inflammatory monocytes (76% CCR2+ of Ly6Chi on average), suggesting increased mobilization of these cells from the bone marrow. A small fraction of Ly6Clo monocytes (7% CCR2+ of Ly6Clo on average) also expressed CCR2, whereas CD11b+ cells that were Ly6C− were also negative for CCR2 (Fig. 7h–j, Supplementary Fig. 5c, d). In A3250 tumors, CCR2 was expressed by a majority of CD11b+ cells (~90%), and in lungs by ~45% of all CD11b+ cells. In tumors, CCR2 was expressed by Ly6Chi monocytes and by Ly6CloF4/80+ macrophages to a comparable extent (Fig. 7h–j and Supplementary Fig. 5e). In the lung, Ly6Chi cells were the main subtype that expressed CCR2, with some expression also by

Ly6Clo macrophages (Fig. 7h–j, Supplementary Fig. 5f). CCR2 was not expressed by CD11b+ F4/80−Ly6C− cells in the blood, tumors or in the lungs (Supplementary Fig. 5c–f). Together, these results demonstrated high levels of CCR2+ inflammatory monocytes in the blood and high CCR2 expression by myeloid cells associated with A3250 tumors and lung metastases. While CCL2 was highly expressed by A3250 cells, other human chemokines that activate CCR2 (CCL7, CCL8, CCL13, and CCL16), were not expressed (Supplementary Fig. 5g, Supplementary Table 6). These findings implicated CCL2 as the key tumor cell-derived agonist of CCR2 in the A3250 model and strongly suggested an important role for CCL2 in monocyte and macrophage recruitment to both A3250 primary and metastatic sites.

**CCL2 knockdown markedly reduces A3250 tumor growth, lung metastasis, and inflammatory phenotype.** To directly investigate the function of CCL2 in the malignant phenotype of A3250 tumors, we performed shRNA knockdown (KD) for CCL2. shCCL2 resulted in a ~63% reduction in CCL2 mRNA, as determined by qPCR, and in a decrease of secreted CCL2 protein in vitro (Fig. 8a, b). CCL2 protein showed multiple bands on Western; including a 11 kDa native form and 13 kDa and 15 kDa glycosylated forms[29]. In tumors, *CCL2* transcripts were reduced

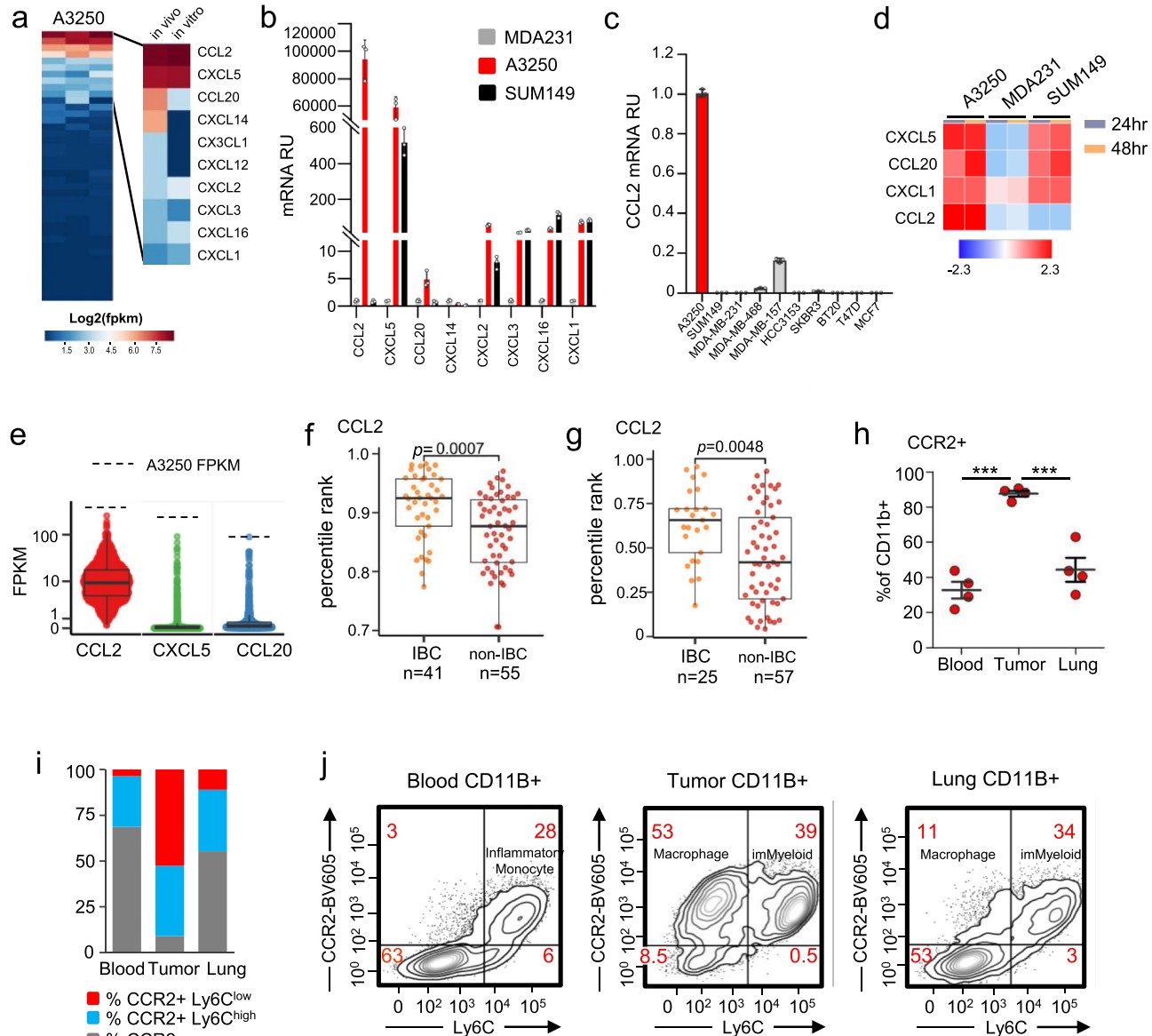

**Fig. 7 Chemokine expression in A3250 and in human IBC. a** mRNA-Seq analysis of human chemokines in A3250 tumors in vivo and cells in vitro, shown as log2(FPKM). **b** qPCR for chemokines as indicated in A3250, SUM149 and MDA-MB-231 cells in vitro. $n = 3$ biologically independent samples. Data are presented as mean values ± SD. Two independent experiments were performed. **c** qPCR for *CCL2* expression in A3250 cells in vitro compared to the SUM149 and eight non-IBC cell lines. $n = 3$ biologically independent samples. Data are presented as mean values ± SD. Two independent experiments were performed. RU = relative units. **d** Chemokines in supernatants of cells as indicated, detected with Olink proteomics. Select chemokines are shown, that were expressed by mRNA and present in Olink Inflammation panel. Heatmap indicates matrix values, data scaled by row. **e** Boxplot showing the distribution of FPKM values for *CCL2, CXCL5*, and *CCL20* chemokines in 1222 TCGA breast cancer samples. Each point represents an individual sample. Dotted line indicates average FPKM for the A3250 tumor ($n = 3$). *CCL2* expression in IBCs and non-IBCs by analysis of human microarray datasets E-MTAB-1006[24] (**f**) and GSE22597[26] (**g**), measured as the percentile rank of signal intensity for the maximally expressed probe for *CCL2*. Box extends to 25th and 75th percentiles, whiskers to the 10–90 percentile and black line denotes the median. The significance of the difference in distributions of percentile ranks in IBC vs non-IBC samples was assessed using Wilcoxon rank sum test. **h–j** Expression of CCR2 on myeloid cell subsets in A3250 tumors, blood and lungs, determined by flow cytometry. Fraction of CCR2+ CD11b+ cells. $n = 4$ biologically independent samples (**h**); fraction of CCR2+ Ly6C$^{hi}$ or CCR2+ Ly6C$^{lo}$ cells (**i**); representative FACS plots of CD11b+ CCR2+ cells in blood, tumors and lungs (**j**). A Type III Analysis of Variance with Satterthwaite's method determined a significant difference between the three groups ($p = 4 \times 10^{-5}$), followed by a Linear Mixed Model fit by REML and t test using Satterthwaite's method with Bonferroni correction (blood, tumor $p = 1.7 \times 10^{-5}$; lung, tumor $p = 6.7 \times 10^{-5}$). Black line on the plot indicates mean ± SEM (**h**) ***$p < 0.001$. FPKM = Fragments Per Kilobase of exon per Million mapped reads. Source data are provided as a Source Data file.

by 76% on average, as determined by mRNA-Seq (Fig. 8c; Supplementary Table 6). Downregulation of CCL2 in A3250 cells did not result in an altered expression of other human CCR2 ligands, which remained undetectable (Supplementary Table 6).

CCL2 KD led to a striking inhibition of tumor growth, as determined by bioluminescent imaging (Fig. 8d). A3250 CCL2

KD tumors also exhibited a striking reduction in clinical signs of skin inflammation characteristic of IBC (Supplementary Fig. 6a–c). Erythema in particular was drastically diminished upon CCL2 KD, and this effect was independent of tumor size. By comparison, SUM149 IBC tumors did not show pronounced erythema (Supplementary Fig. 6a, b). Furthermore, CCL2 KD

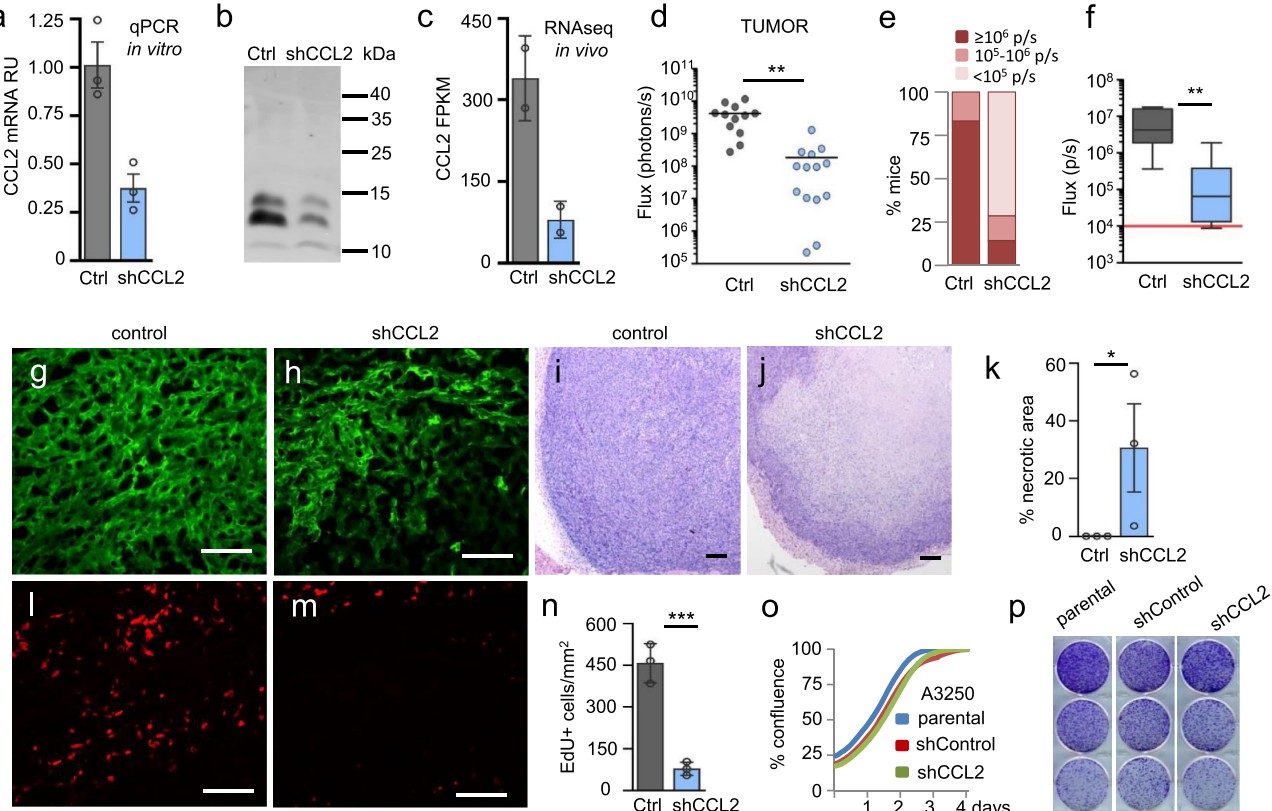

**Fig. 8 Effects of CCL2 knockdown on A3250 tumor growth and metastasis. a** *CCL2* mRNA expression levels in A3250 control and A3250-shCCL2 cells determined by qPCR. $n = 3$ biologically independent samples. Data are presented as mean values ± SD. Two independent experiments were performed. **b** CCL2 Western analysis of A3250 cells in vitro as indicated. **c** *CCL2* mRNA levels in A3250-control and A3250-shCCL2 tumors determined by mRNA-Seq. $n = 2$ biologically independent samples. Data are presented as mean values ± SD. Two independent experiments were performed. **d** Bioluminescence imaging (total Flux) of A3250-control ($n = 12$ tumors from 6 mice) and A3250-shCCL2 tumors ($n = 14$ tumors from 7 mice) in vivo 6 weeks post-injection. Two tumors were harvested from each mouse, therefore to control for repeated measures, a Linear Mixed Model fit by REML and $t$ test using Satterthwaite's method was used for analysis ($p = 0.00115$). **e** Metastatic burden in the lungs of A3250-control ($n = 6$) and A3250-shCCL2-bearing ($n = 7$) tumors determined by ex vivo bioluminescence imaging at 6 weeks. Signal intensity is indicated in photons/second (p/s). Incidence of metastases: macro-metastases, $\geq 10^6$ p/s; micro-metastases, $10^5$–$10^6$ p/s; disseminated tumor cells, $10^4$–$10^5$ p/s. **f** Box and whisker plot showing total metastatic burden by bioluminescence in a mouse cohort as indicated ($n = 6$ A3250-control and $n = 7$ A3250-shCCL2 biologically independent lung samples). Solid black lines denote mean values. Red solid line indicates the threshold of positive Fluc signal above background. A two-tailed Mann–Whitney $t$ test yielded $p = 0.005$. **g, h** Viable tumor area of control and A3250-shCCL2 tumor sections visualized by GFP. **i, j** H&E staining showing viable and necrotic tumor areas in tumors as indicated. **k** Quantification of necrotic tumor areas based on H&E staining ($n = 3$ biologically independent tumors per cell line). Data are presented as mean values ± SEM. $P = 0.0453$, determined through a one sample two-tailed $t$ test. **l, m** EdU labeled proliferating cells (red) in sections of control and shCCL2 tumors shown in (**g**) and (**h**). **n** Proliferation of tumor cells in vivo, determined by EdU quantification on GFP+ tumor areas. $n = 3$ biologically independent tumors per cell line. Data is presented as mean values ± SD. A two-tailed unpaired $t$ test yielded $p = 0.0009$. Images are representative of two independent experiments. **o** Proliferation of tumor cells in vitro. $n = 3$ biologically independent samples per condition. Data are representative of two independent experiments. **p** Colony Forming Assay on plastic. $n = 3$ biologically independent samples per condition. Images are representative of two independent experiments. All scale bars 100 μm. **p < 0.05 **p < 0.01 ***p < 0.001. RU = Relative Units; FPKM = Fragments Per Kilobase of exon per Million mapped reads. Source data are provided as a Source Data file.

decreased both the incidence and the overall metastatic burden of lung metastases (Fig. 8e, f). The majority of mice bearing CCL2 KD tumors lacked macro-metastases in the lungs, and mainly presented with disseminated tumor cells, whereas mice with control A3250 tumors mainly presented with macro-metastases. Histopathological analysis of cytokeratin-stained tumor sections revealed that the invasiveness of A3250 tumors was not decreased upon CCL2 KD (Supplementary Fig. 7a–f), suggesting that CCL2 KD primarily decreased metastatic colonization rather than dissemination.

Primary tumors exhibited an early onset of tumor necrosis upon CCL2 KD. Analysis of endogenous expression of GFP in tumor sections revealed that in CCL2 KD tumors, only the peripheral tumor area was viable (Fig. 8g, h), and H&E staining showed large necrotic areas in comparison to control tumors at

the four-week time-point (Fig. 8i–k). This was not due to the inhibition of angiogenesis, since blood vessel densities in tumors were not changed upon CCL2 KD (Supplementary Fig. 7g–i). CCL2 KD also reduced tumor proliferation by as much as 83%, with an average number of EdU+ cells/mm² tumor decreased from 458 to 78 per 6 h (Fig. 8l–n, Supplementary Table 3). In contrast, CCL2 KD had no effect on proliferation and colony formation of tumor cells in vitro (Fig. 8o, p), arguing that CCL2 did not stimulate A3250 growth in an autocrine manner. Similar in vitro and in vivo data were obtained using A3250 cells transduced with an independent shCCL2 construct (Supplementary Fig. 8). In accordance, A3250 tumor cells lacked detectable levels of the *CCR2* receptor by mRNA-Seq (Supplementary Table 7). In contrast, *CCR2* was highly expressed by mouse cells in the microenvironment of A3250 tumors (Supplementary

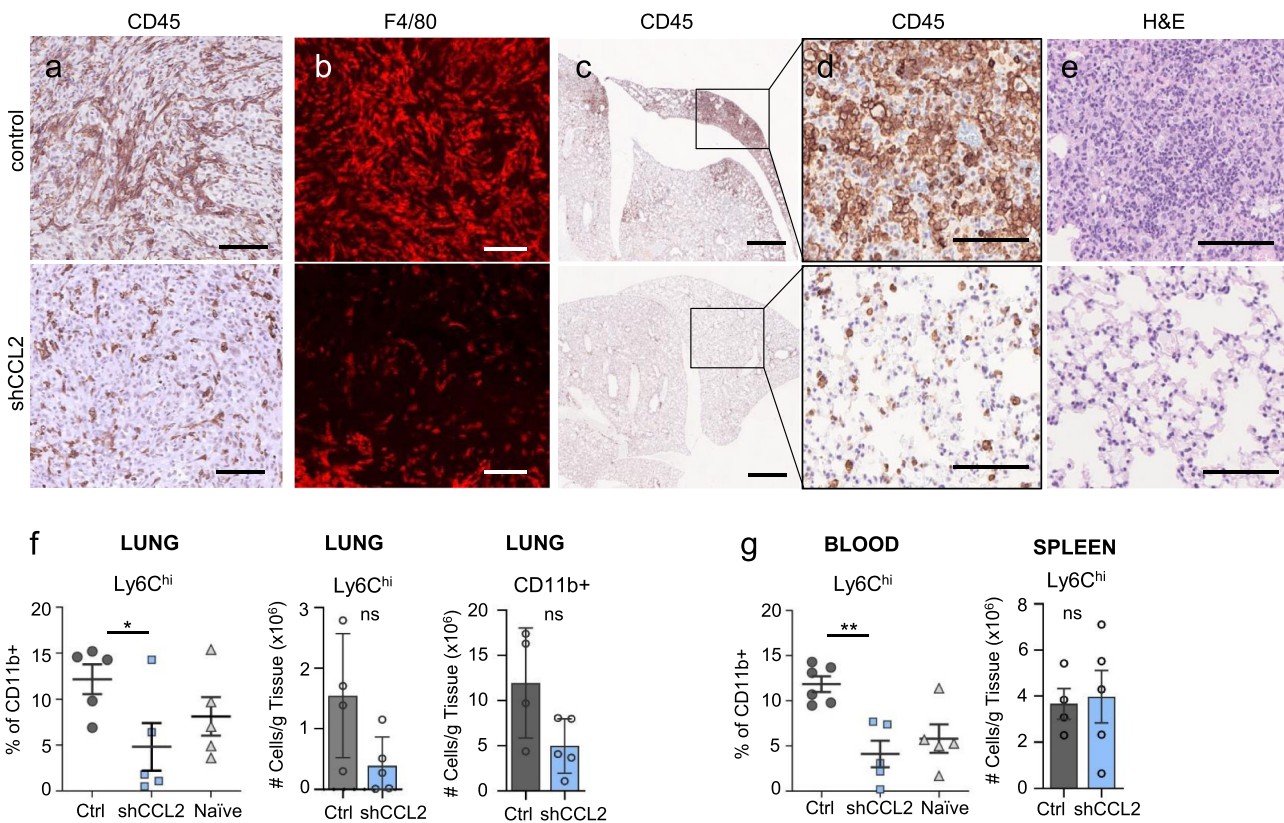

**Fig. 9 Local and systemic effects of CCL2 knockdown in A3250 tumors.** Immunostaining of A3250-control and A3250-shCCL2 tumors for CD45 (**a**) and F4/80 (**b**) as indicated. **c–e** Lungs of mice bearing A3250-control or A3250-CCL2 tumors at 4 weeks post-injection (~40 mm³ volume), immuno-stained for CD45 (**c, d**) or stained for H&E (**e**). Flow cytometry analysis of CD11b+ and Ly6Chi cells in lungs (**f**), blood and spleen (**g**) from A3250 tumor-bearing (control), A3250-shCCL2 and naïve mice. **f** left panel, $n = 5$ biologically independent samples for each group, $p = 0.0158$; **f** middle and right panels, $n = 4$ A3250-control and $n = 5$ A3250-shCCL2 biologically independent lung samples. **g** left panel, $n = 6$ A3250-control and $n = 5$ for A3250-shCCL2 and tumor-naïve independent blood samples, $p = 0.0011$; (**g**) right panel, $n = 4$ A3250-control and $n = 5$ A3250-shCCL2 biologically independent spleen samples. Black lines on the plots indicate mean values ± SEM. (**f-g**), left panels: performed a two-sample $t$ test to determine significance; (**f**), center and (**f-g**), right panels: significance was determined through a Mann–Whitney non-parametric $t$ test. Images are representative of two independent experiments. $n = 3$ biologically independent mice. All scale bars: 100 µm, except (**c**), 1 mm. **$p < 0.01$, *$p < 0.05$; ns, not significant. Source data are provided as a Source Data file.

Table 7). These results indicated that CCL2 did not promote A3250 tumor growth through autocrine stimulation and supported the concept that macrophages recruited by CCL2 promoted A3250 tumor cell proliferation and viability in vivo.

**CCL2 downregulation in tumors decreases macrophage infiltration and systemic inflammation.** Immunostaining of immune cells in CCL2 KD tumors revealed a striking reduction of CD45+ cells in viable tumor areas, most of which were F4/80+ macrophages (Fig. 9a, b), thus correlating the loss of tumor macrophages with inhibition of tumor growth. To examine the systemic effects of CCL2 downregulation, we examined the lungs of A3250 tumor-bearing mice during the early stages of tumor dissemination by immunostaining (3–4 weeks post-injection). In the lungs of mice bearing A3250 control tumors, hotspots of dense inflammatory infiltrates associated with disseminated tumor cells were observed (Fig. 9c–e). These inflammatory hotspots were not present in the lungs of mice bearing A3250/CCL2 KD tumors, and CCL2 KD led to a striking normalization of the lung architecture (Fig. 9c–e). Flow cytometry revealed a reduction of Ly6Chi cells in the lungs by 75%, to the normal level, and of total CD11b+ cells by 58% (Fig. 9f). These results correlated with a decrease of Ly6Chi inflammatory monocytes in the blood to the normal level of naïve age-matched mice (88% reduction) (Fig. 9g). In contrast, the number of Ly6Chi cells in the spleen was unchanged

upon CCL2 KD and was not correlated with the levels of Ly6Chi cells in the blood or in the lungs (Fig. 9g). This decrease in inflammatory cell accumulation in the lungs correlated with the decrease in A3250 lung metastases, consistent with a metastasis-promoting role of Ly6C+ cells in the lung.

**Human IBCs are enriched in macrophages.** A recent study identified distinct clusters of human TNBCs based on immune infiltrates that ranged from macrophage-enriched to neutrophil-enriched[21]. To examine whether human IBCs are macrophage-enriched, we analyzed microarray data sets of 41 human IBCs and 55 non-IBCs[24,25]. We applied a 37-gene signature specific for TAMs in breast cancer[30]. Hierarchical clustering revealed two groups of breast cancers, enriched or not enriched in TAM genes. Among breast cancers, there were significantly more IBC samples enriched in TAM genes (60% IBCs, 40% non-IBCs), than not enriched in TAM genes (24.5% IBCs, 75.5% non-IBCs) ($p$ value of the chi-square test of 0.007) (Fig. 10a).

We extended this analysis by immunostaining an independent cohort of nine human IBCs for the presence of macrophages (CD68+) and neutrophils (CD66b+) (Fig. 10b and Supplementary Fig. 9). High macrophage content was seen throughout the tumors, both in the stroma and/or within tumor cell clusters, in tight apposition to tumor cells. In IBC cases presenting as tumor cell clusters in dermal lymphatic vessels, we observed a dense

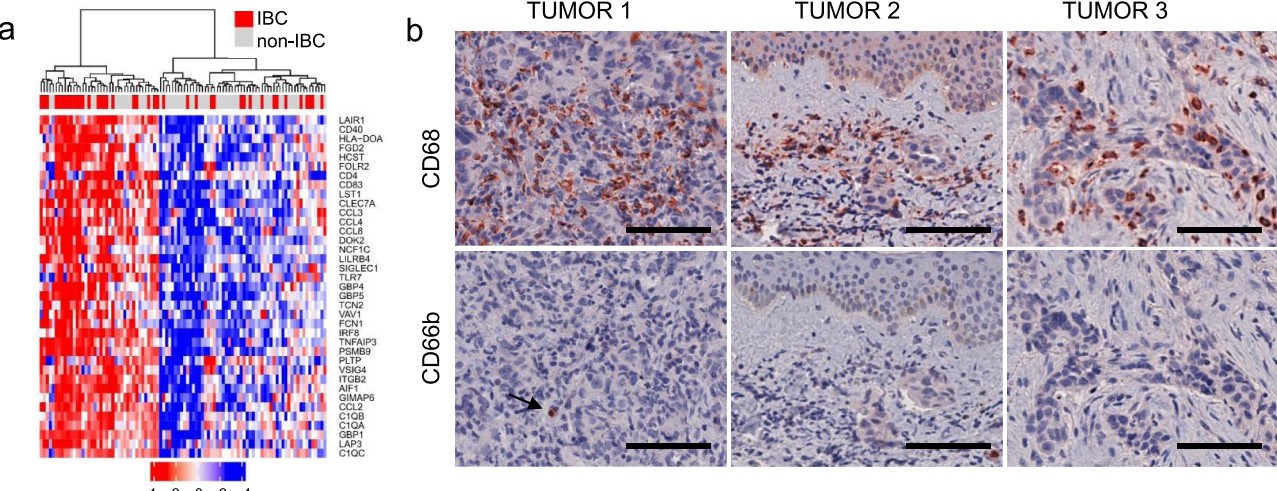

**Fig. 10 Macrophages and neutrophils in human IBC. a** Hierarchical clustering of human IBCs and non-IBCs based on expression of genes characteristic for tumor-associated macrophages (TAMs) in breast cancer. TAM-specific 37-gene signature[30] was applied to microarray data from human IBCs ($n = 41$) and non-IBCs ($n = 55$)[24]. Percentile rank of gcrma normalized microarray signal intensities was z-score transformed and z-scores are shown. **b** Human IBC samples immunostained with antibodies against CD68 for macrophages and CD66b for neutrophils, as indicated. Images shown are representative of $n = 9$ tumors with different clinical presentations of IBC: IBC presenting as a solid tumor (tumor 1), as a lymphovascular emboli (tumor 2) or with highly diffuse pattern of growth (tumor 3). Scale bars: 100 μm.

accumulation of CD68+ macrophages in the dermis. In stark contrast, neutrophils were rare in all IBC tumors analyzed (Fig. 10b, Supplementary Fig. 8). These results were consistent with the gene expression analysis indicating that human IBCs are macrophage enriched, and in agreement with our findings with A3250 tumors, further underscoring the relevance of the A3250 tumor model to human IBC.

## Discussion

IBC remains a major clinical challenge, as this highly aggressive form of breast cancer is the most difficult to diagnose and treat, yet has no specific targeted therapies[31]. Understanding of the cellular and molecular determinants of IBC has lagged far behind that of other breast cancers, largely due to the lack of adequate models for this disease and the relative rarity of this tumor phenotype. Although several human IBC-derived cell lines have been reported[9–11,13], none recapitulates the full spectrum of IBC features in a mouse model. The human triple-negative IBC cell line A3250, described here, closely recapitulates the key features of human IBC in a mouse xenograft model, including skin erythema, flat tumor appearance, diffuse tumor growth pattern, high stroma to tumor ratio, lympho-vascular tumor emboli in the skin, and extensive lymph node and distant metastases. The fact that the SCID mouse strain utilized is deficient in adaptive immunity indicates that T and B lymphocytes are not necessary for the aggressive IBC phenotypic manifestations.

It has been generally assumed that the inflammatory phenotype of IBC, which includes erythema and edema of the breast and overlying skin[32,33], is a consequence of tumor emboli in dermal lymphatics, which are believed to impede egress of inflammatory cells from the tissue[4]. While dermal lymphatic vessel invasion and clinical symptoms of inflammation coincide in IBC, there have been no studies to date concerning their causal relationship. We demonstrated in the A3250 IBC tumor model that inflammation is driven directly by the tumor, and is not secondary to dermal lymphatic vessel invasion. This conclusion is based on our observations from the longitudinal analysis of tumor development that demonstrated vigorous recruitment of inflammatory cells into the tumor and adjacent skin early in tumor development, prior to dermal vessel invasion and

formation of lympho-vascular emboli. This does not preclude, however, that at the later stages of tumor development potential changes in lymphatic vessel functions may further impact tissue inflammation.

IBC is still primarily a clinical diagnosis, and despite the inflamed clinical appearance of the skin, very few studies have investigated inflammatory cells in IBC[34–36], and to what extent they contribute to IBC progression[33]. We observed exceptionally high macrophage content in A3250 tumors and in adjacent skin, and this was confirmed in human IBC tissues. Macrophages were frequently seen in direct contact with IBC tumor cells, whereas in most breast cancers macrophages are typically localized at the tumor periphery or in the stroma. High macrophage densities were also observed in uninvolved skin, in line with a recent report that in patients, breast tissue not involved with IBC was enriched in macrophages[37]. Another study found a significant increase of CD14+ monocytes in blood and in tumors of patients with IBC compared to non-IBC[38], consistent with our findings. Taken together, our data reveal that the prominent presence of macrophages in close contact with tumor cells is a common characteristic of IBC. Finally, in most human breast cancers and in mouse models, neutrophils are the most abundant subset of myeloid cells[21,39,40], while in A3250 tumors and in human IBCs neutrophils were sparse. A recent study of immune cell profiles in breast cancer categorized TNBCs based on TAN and TAM frequency[21]. Accordingly, IBC can be classified as a macrophage-enriched subtype with low TAN to TAM ratio.

CCL2 is a well-known mediator of the innate response and inflammation that supports breast cancer progression through several mechanisms[27,28,41]. TAMs recruited by CCL2 facilitate early steps of metastases by promoting tumor cell migration and intravasation[42,43]. CCL2 enhances recruitment and retention of metastasis-associated macrophages[27] and has been shown to promote metastatic recurrence after chemotherapy[39,44]. We demonstrated that the high level of endogenous CCL2 expression by A3250 tumor cells was a key factor in driving the inflammatory cell influx in IBC and identified the important function of CCL2 in promoting IBC tumor growth. CCL2 KD, and consequently reduction in TAM densities, led to a striking reduction of tumor cell proliferation and a concomitant increase in tumor

necrosis. Thus, A3250 cells were highly dependent on TAMs for primary tumor growth, while exaggerated systemic inflammation driven by exceptionally high levels of CCL2 promoted expansion of metastases. These results make it tempting to speculate that macrophages may directly support tumor cell metabolic needs critical for proliferation and survival. Of note, SUM149, an IBC line, which does not express CCL2 at the RNA and protein level, showed a much less aggressive IBC phenotype, lacking erythema or dermal lymphatic invasion and with significantly less inflammatory infiltrate as compared to A3250. These findings further support the important role of high level endogenous CCL2 production in promoting the florid IBC phenotype. Finally, our evidence that macrophages are a dominant subset in clinical IBCs suggest that immunotherapeutic strategies that deplete or otherwise target macrophages may be promising therapeutic approaches for IBC. Thus, this IBC tumor model provides a discovery tool with which to facilitate further understanding of IBC biology and potentially foster the development of much needed IBC-targeted therapeutics.

## Methods

**Cells and culture conditions**. The A3250 breast carcinoma line was isolated from a primary tumor of a 50-year-old anonymized patient with IBC, provided to the Aaronson lab from a tissue biorepository directed by Dr. Dirk Iglehart, Duke University. We have complied with all relevant ethical regulations. Duke IRB Pro00012025 "Breast Cancer Blood/Tissue Repository and Database" is a continuation of Dr. Iglehart's IRB protocol permitting them to provide anonymized human tumor tissues as byproducts of standard clinical procedures to other investigators for cancer research. The Mount Sinai IRB considers research performed with anonymized human samples as non-human subjects research. The A3250 IBC cell line was established in culture by the Aaronson lab as performed for other A series tumor lines[15,45,46]. Briefly, the tumor was cut into 2 mm pieces in high-glucose DMEM supplemented with 4 mM L-glutamine (Sigma, Cat. # D6429), 10% FBS (Corning, Cat. # 35-010-CV) and 1X antibiotics (100 U/ml penicillin, 100 U/ml streptomycin) (Gibco, Cat. # 15140-122), and pieces were evenly dispersed within several T25 tissue culture flasks (~12 pieces/flask). Cultures were incubated at 37 °C and medium was replaced only if pH changes occurred or when significant debris was present. Once confluent, cells were split at a 1:2 ratio.

The SUM149 IBC and HCC3153 TNBC cell lines were generously provided by Dr. Stephen Ethier (Medical University of South Carolina), and Dr. John Minna, UT Southwestern[47], respectively. MDA-MB-157, MDA-MB-231, SKBR3 and MCF7 breast cancer cell lines were obtained from ATCC.

A3250 and all other tumor cells except MDA-MB-231 and SUM149 were cultured in high-glucose DMEM supplemented with 4 mM L-glutamine (Sigma, Cat. # D6429), 10% FBS (Corning, Cat. # 35-010-CV) and 1X antibiotics (100 U/ml penicillin, 100 U/ml streptomycin) (Gibco, Cat. # 15140-122). MDA-MB-231 cells were cultured in RPMI 1640 (Gibco, Cat. No. 11875119) supplemented with 2 μg/ml insulin, 4 mM L-glutamine, 10% FBS and 1X antibiotics as above. SUM149 cells were cultured in Ham's F12 supplemented with 5 μg/ml insulin, 1 μg/ml hydrocortisone, 4 mM L-glutamine, 5% FBS and 1X antibiotics. All cell lines were confirmed mycoplasma-free prior to use.

**Retrovirus and lentivirus generation**. To generate retroviruses for constitutive gene expression, the expression plasmid of interest was co-transfected with the pCL-ampho packaging plasmid in HEK293T cells. For lentivirus production[46], HEK293T cells were co-transfected with expression plasmid of interest, pCMV Δ8.91 packaging and pMD VSV-G envelope plasmids using Lipofectamine 2000 according to manufacturer's instructions. Conditioned medium containing viral particles was harvested two, three and four days after transfection and virus stock was pooled and aliquots were stored at −80 °C until use.

**Generation of stable cell lines**. To generate A3250 and MDA-MB-231 cells expressing eGFP (enhanced Green Fluorescent Protein) and firefly luciferase (Fluc) (A3250-eGFP-FLuc and MDA-MB-231-eGFP-Fluc), parental cells were transduced with the pCCBS-EGFP retrovirus, followed by puromycin selection[46]. The cells were then transduced with the NSNI-luc lentivirus[45] and selected with neomycin and puromycin.

A3250-shCtrl and A3250-shCCL2 cells were generated by sequential lentiviral infection of A3250-eGFP-FLuc cells with either VIRHD-bla empty vector or with two VIRHD-bla vectors containing CCL2 targeting sequences GCTGTTATAACTTCACCAATA and TCATAGCAGCCACCTTCATTC, followed by selection with blasticidin, neomycin and puromycin. Independently, A3250-shCtrl#2 and A3250-shCCL2#2 cells were generated by transducing A3250-eGFP-Fluc cells with either pLKO.1 scramble or pLKO.1 vector with CCL2 targeting sequence GATGTGAAACATTATGCCTTA from Sigma MISSION

shRNA (Drug Discovery Institute, Mount Sinai). Cells were maintained in the presence of puromycin and neomycin. All transductions were performed in presence of 8 μg/ml polybrene.

**Cell line authentication**. Short tandem repeat (STR) data were generated on the SeqStudio Genetic Analyzer, at the Dept. of Oncological Sciences Sequencing core. We confirmed that the STR signatures of A3250 cells did not match STRs of known cell lines in the ATCC or DSMZ STR database. SUM149, MDA-MB-157, MDA-MB-231, HCC3153, SKBR3, and MCF7 cell lines were also authenticated using STR analysis. Cells were prepared per the manufacturer's recommendations (FTA Sample Collection Kit, cat. no. 135-XV) and processed as described[48].

**IncuCyte proliferation assay**. A3250 cells were seeded to 40% confluence in 6-well plates in complete media with 10% FBS, allowed to adhere for 6 hours, and subsequently changed to media containing 10% or 1% FBS. Cells were imaged every four hours for four days using the Incucyte™ ZOOM (Essen BioScience Inc, GUI version 2018), for a total of 16 10X images per well for each time-point. Media was changed once, 48 hr after the start of imaging. Average percent confluence was calculated for each well at every time point ($n = 2$ wells per condition).

**Colony formation assay**. A3250 and MDA-MB-231 tumor cells were seeded in triplicate into 6-well plates at a density of 500 or 250 cells per well (52 or 26 cells/cm²). Cells were seeded in complete media with either 10% or 1% FBS and incubated for 11 days, with media changes every four days. Cells were then fixed with ice-cold methanol (100%) for 10 min and stained with a 0.2 % crystal violet in methanol for 30 min at room temperature. Images of each well were taken and colony numbers were quantified using ImageJ v. 1.52j.

**Mouse experiments and tissue collection**. SCID mice were used in all experiments unless indicated otherwise. Animals were housed in climate-controlled barrier conditions with automated light/dark cycles. A3250 ($1 \times 10^6$ cells in 100 μl PBS, Phosphate-Buffered Saline solution), SUM149 ($2 \times 10^6$ cells in 100 μl PBS) or MDA-MB-231 ($2 \times 10^6$ cells in 100 μl PBS) tumor cells were injected orthotopically into the fourth mammary fat pads on both sides of six to eight-week-old SCID or NOD SCID mice (Charles River Laboratories). For most experiments, SCID mice were bred inhouse. Mouse weights and tumor sizes were measured weekly. Mice were sacrificed one to eight weeks post-injection, and primary tumors and organs were collected. Six hours before sacrifice, mice were injected with 100 μg of 5-ethynyl-2′-deoxyuridine (EdU). For each mouse, tumors, sentinel lymph nodes (inguinal), distant lymph nodes (axillary and brachial), lungs and liver were collected. Tumors and organs were fresh-frozen in OCT, or fixed 1 hr in formalin and embedded in paraffin. Organs collected from mice with GFP-labeled tumors were fixed for 1 hr in formalin and frozen in OCT, to preserve GFP and allow the evaluation of metastases at a cellular resolution. The percent change in body weight was determined through the difference between the body weights measured during week 1 and week 6 (i.e. the endpoint) of the experiment. Mouse experiments were approved by the Icahn School of Medicine at Mount Sinai's Institutional Animal Care and Use Committee (IACUC) and in compliance with all relevant ethical regulations for animal testing and research. Maximal tumor burden permitted by Mount Sinai's IACUC is 10% of mouse body weight or tumor diameter of 10 mm in any dimension. We confirm that tumors did not exceed maximum approved tumor size or burden. In most experiments tumors were collected when 5–7 mm in diameter.

**Bioluminescent imaging of tumors and metastases**. Mice were anaesthetized and background luminescence signals were acquired. Mice then received an intraperitoneal injection of D-luciferin dissolved in sterile PBS (150 mg/kg), and images were acquired every two minutes until peak luciferin signal was achieved for each mouse. Subsequent to in vivo imaging, mice were sacrificed and organs were quickly harvested. D-luciferin was added again to excised organs and metastases in organs were imaged ex vivo. Organs from a mouse injected with the parental tumor cell line not labeled with Fluc were imaged ex vivo and used to determine background values for each tissue. Bioluminescent imaging was performed on the IVIS® Spectrum (PerkinElmer) and analyzed using LivingImage software, version 4.5 (PerkinElmer).

**Visualization of metastases by GFP in whole organs and in tissues sections**. To evaluate macro-metastases, formalin-fixed whole organs from mice bearing eGFP-Fluc expressing tumors were imaged with the Leica MZ16 FA stereomicroscope and DFC300 FX camera before embedding in OCT. Lymph node and lung metastases were evaluated at single-cell resolution in frozen sections (6 μm and 10 μm, respectively) of formalin-fixed tissues. Each section was mounted on a microscope slide, rehydrated briefly in PBS, and coverslipped with Vectashield HardSet mounting medium with DAPI (Vector Laboratories H-1500). Sections were immediately analyzed for eGFP signal using the Nikon Eclipse E600 microscope and DS-Qi1 monochrome camera. The number of sections analyzed for each lymph node was 4-16, and for each lung was 16–27. Metastases were categorized as disseminated single tumor cells, micrometastases (clusters of 2–10 tumor cells) or

macrometastases (clusters of >10 tumor cells). The bioluminescence value for each tissue was then compared to the metastasis evaluation by GFP, and the following classification emerged: tissues with Fluc values $<10^4$ contained no metastases detectable by GFP analysis; Fluc values ranging from $\geq 10^4$ to $<10^5$ represented few isolated tumor cells and some micrometastases; Fluc values from $\geq 10^5$ to $<10^6$ reflected mainly micrometastases and some isolated tumor cells; and Fluc values $\geq 10^6$ reflected presence of macrometastases, in addition to many micrometastases and isolated tumor cells.

**Immunofluorescent staining and quantification**. The following antibodies were used for immunostaining of frozen tissue: LYVE-1 (Abcam ab14917; 1:600), CD34 (Invitrogen 14-0341-82; 1:100), CD45 (Abcam ab10558; 1:100), F4/80 (Bio-Rad MCA497; 1:100), CD11b (Biolegend 101201; 1:200), Ly6G (Biolegend 127601; 1:100) and pan-cytokeratin (Abcam ab9377; 1:100). Tissues were incubated in 100% acetone ($-20\,°C$, 2 min) and 80% methanol ($4\,°C$, 5 min), primary antibodies were diluted in 12% BSA/PBS, and incubated for 1 hr (LYVE-1, CD45, Ly6G) or 2 hr (F4/80, CD11b, pan-cytokeratin) in a humidity chamber at room temperature. Corresponding secondary antibodies labeled with Alexa Fluor 488 (Invitrogen cat. no. A-11070), 555 (Invitrogen cat. no. A-21434) or 594 (Invitrogen cat. no. A-11037) were diluted to 1:300 and incubated at room temperature for 30–45 min. Nuclei were counterstained with DAPI (10 μg/ml). Slides were coverslipped using fluorescence mounting medium (Dako cat. no. S3023). Images were captured using the Nikon Eclipse E600 microscope and DS-Qi1 monochrome camera. Whole-slide scans were acquired with the NanoZoomer S60 scanner C13210-01 (Hamamatsu).

Angiogenesis was quantified by measuring the area of CD34+/LYVE-1- vessels in tumor sections using NIS Elements 3.22 (Nikon). Three 20X images per tumor were quantified (total area of $2.5 \times 10^6$ μm$^2$ per tumor, $n = 3$ tumors per experimental condition) and data expressed as vessel area/total tumor area analyzed (% vessel area).

TANs and TAMs were quantified by measuring Ly6G+ and F4/80+ cell area, respectively, in tumor sections using NIS Elements 3.22 (Nikon). One entire scanned section of each tumor was quantified (total area ranged from 1–3.42 mm$^2$ per tumor, $n = 3$ tumors) and data expressed as % positive cell area/ viable tumor area analyzed.

**EdU proliferation and GFP quantification in tumors**. EdU was visualized in formalin-fixed OCT-embedded frozen tumor sections using the Click-iT™ EdU Imaging Kit with Alexa Fluor 594 (red) (Invitrogen cat. no. A-11037, 1:300). Following EdU staining, sections were incubated in 0.5% TritonX/PBS at room temperature for 20 min, and immunostained with anti-GFP antibody (Abcam ab6556, 1:100) to enhance the GFP signal. Images were captured with the Nikon Eclipse E600 microscope and DS-Qi1 monochrome camera. Tumor cell proliferation was determined by quantifying EdU+ signal in the GFP-positive tumor areas using NIS Elements 3.22 (Nikon). Three 20x images were quantified per tumor (total area of $2.5 \times 10^6$ μm$^2$ per tumor, $n = 3$ tumors per experimental condition). Data were expressed as number of EdU+ cells/mm$^2$ GFP+ tumor area. To calculate tumor-stroma ratio, GFP images were overlaid onto the corresponding DAPI images to determine tumor boundaries and exclude necrotic areas. GFP+ viable tumor area was quantified as above and data expressed as GFP+ tumor area/ total tumor area analyzed (% GFP + area).

**Macrophage proliferation in tissues**. EdU was visualized in formalin-fixed OCT-embedded frozen tumor sections using the Click-iT™ EdU Imaging Kit with Alexa Fluor 488 green label (Invitrogen cat. no. A-11070, 1:300). Following EdU staining, sections were incubated in 0.5% TritonX/PBS at room temperature for 20 min and immunostained with F4/80 antibody (Bio-Rad MCA497; 1:100) as described above.

**Optical clearing and 3D reconstruction of tumors**. A3250-eGFP tumors were fixed in formalin for 1 hr, embedded in OCT and frozen. 300 μm thick sections were optically cleared using the Ce3D method[49]. Images were captured with a Zeiss LSM 880 Airyscan confocal microscope using 20x objective. Images 512 × 512 pixels were acquired at 1X optical zoom, with a voxel size of 3 μm depth. A z-stack image was then exported and processed using Imaris software (Oxford Instruments) to generate a 3D image.

**Immunohistochemistry**. Immunohistochemistry was performed on paraffin-embedded tissue sections using the Leica Bond RX automated immunostainer (Leica Biosystems), according to the Leica staining protocol. Briefly, all slides were deparaffinized using a heated Bond™ Dewax Solution (Leica cat. no. AR9222) and washed with 1X Bond™ Wash Solution (Leica cat. no. AR9590). Epitope retrievals were carried out for 20 minutes using the citrate-based Bond™ Epitope Retrieval 1 solution (Leica cat. no. AR9961). Slides were then blocked with either the Dako Dual Endogenous Enzyme Block (Agilent S2003) for ten minutes prior to staining with the Bond™ Polymer Refine Red Detection System (Leica cat. no. DS9390), or with a 3–4% v/v hydrogen peroxide block for five minutes that is incorporated into the Bond™ Polymer Refine Detection kit (Leica cat. no. DS9800) and used for DAB staining. All antibodies were diluted using Bond™ Primary Antibody Diluent (Leica Biosystems AR9352).

For tumor xenografts, antibodies against Ki-67 (Leica Biosystems Novocastra NCL-Ki67p; 1:1000), pan-cytokeratin (Abcam ab9377; 1:100), CD45 (Abcam ab10558; 1:1000), Iba1 (Wako 019-19741; 1:1000), ErbB2 (Abcam ab134182; 1:100), estrogen receptor (BioGenex NU710-UC; 1:50), and progesterone receptor (Abcam ab16661; 1:400) were incubated for one hour at room temperature, and the following detection kits were used: Refine red for Ki-67 and cytokeratin, and DAB for CD45, Iba1, ErbB2, estrogen receptor, and progesterone receptor staining.

Anonymized human IBC tumor samples, formalin-fixed and paraffin-embedded, were obtained from the Mount Sinai pathology archive, under the auspices of the Department of Pathology and Tisch Cancer Institute Bioresository Core, and were independently evaluated by another pathologist. According to ISMMS IRB criteria, human tissue samples collected as byproducts of standard care and anonymized are considered non-human subjects research. Out of nine samples, six were tumors, assessed as highly invasive ductal carcinomas, whereas three samples presented exclusively as tumor emboli in dermal lymphatics. Human IBC tumor samples were immunostained with anti-CD66b (BD Biosciences 555723; 1:600) and anti-CD68 antibodies (Dako M0814; 1:100) on the same sections, using Leica Bond RX immunostainer and according to the protocol for Multiplexed Immunohistochemical Consecutive Staining in Single Slide (MICSSS)[50,51]. Whole-slide scans were acquired with the NanoZoomer S60 scanner C13210-01 (Hamamatsu). Alternatively, images were captured with the Nikon Eclipse E600 microscope and DS-Ri1 color camera.

**Flow cytometry**. Xenograft mouse tumors ~5 mm in diameter, spleens and lungs were minced into 1 mm$^3$ pieces. Age and sex-matched tumor naïve mice were used as controls. Tissues were digested in collagenase D (Sigma 11088858001) using the MACSmix tube rotator (Miltenyi) at $37\,°C$, and pressed through a 70 μm filter. Lymph nodes were dissociated by mechanical disruption. Blood was collected by cardiac puncture. Erythrocytes in blood, spleen and lung were lysed using RBC lysis buffer (eBioscience 00-4300-54), as per the manufacturer's protocol. Cells were counted using the Countess II Automated Cell Counter (Invitrogen). Samples were stained with antibodies on ice for 30 minutes in FBS Stain Buffer (BD Pharmigen 554656), and run on the BD LSRFortessa X-20 cell analyzer. The antibodies used were all from Biolegend: anti-mouse CD45-BV785 (Biolegend 103149), CD11b-APC (Biolegend 101212), Ly6G-PE (Biolegend 127608), Ly6C-BV421 (128032), F4/80-FITC (123108), and CCR2-BV605 (Biolegend 150615) (all at 1 μg/ml). Dead cells were excluded from the analysis based on DAPI signal (50 ng/ml). Analysis was performed using BD FACSDiva™ software (v.8.0.1) (BD Biosciences). Contour plot and histogram images were generated using FCS Express version 6.0 (De Novo Software). Absolute cell numbers were calculated using AccuCheck Counting Beads (Invitrogen PCB100). Cells were identified on FSC-A/SSC-A plot and doublets were excluded using FSC-A/FSC-H plot. Viable cells were identified using FSC-A/DAPI plot and CD45+ cells were identified on FSC-A/CD45 plot. CD11B+ myeloid cells were identified on CD45/CD11B plot. Boundaries between positive and negative staining cell populations were defined using Fluorescence Minus One controls.

**Western Blotting**. For CCL2 detection, A3250-control or A3250-shCCL2 tumor cells (~$7 \times 10^6$ cells) were cultured in 5 ml serum-free media for 24 hr. Conditioned media were concentrated 23-fold using Amicon Ultra Centrifugal Filter Units (Ultracel-3K, Millipore) and 20 μl was loaded on a 12% SDS-PAGE gel. Western blot analysis[52] was performed[52] using anti-human CCL2 (R&D Systems MAB679) at 1 μg/ml. Secondary antibody, Alexa Fluor 680 goat anti-mouse IgG (Invitrogen, A-21059) was used at 1:10,000 dilution and signal was detected using the LI-COR Odyssey fluorescence imaging system. Western blot source data is provided as a Source Data file.

**Olink multiplex immunoassay**. Supernatants of A3250, SUM149 and MDA-MB-231 cultures were collected at 24 h and 48 h after seeding $1 \times 10^6$ cells per 10 cm plate in 8 ml of standard media for each cell line. The conditioned media was harvested and analyzed for chemokines within the Olink inflammation panel (Olink Bioscience, Sweden) by the Human Immune Monitoring Center, Mount Sinai. The assay uses proximity extension assay (PEA) technology wherein oligonucleotide labeled antibody pairs allow pair wise binding to target proteins. The antibody bound target antigens which correspond to oligonucleotides from an amplicon are then quantified by high throughput qPCR and provided as relative quantification of protein concentration denoted as Olink proteomics arbitrary unit NPX (normalized protein expression) on a log2 scale.

**RNA Isolation and qPCR**. Total RNA was isolated from subconfluent cultures after 24 hr of plating. RNA was isolated using the RNeasy kit (Qiagen), RNA concentration was estimated using a NanoDrop 1000 spectrophotometer (Thermo Scientific) and cDNA was generated from 1 μg RNA using SuperScript II reverse transcriptase (Invitrogen). qPCR was performed with ViiA Real Time PCR System (Life Technologies, Carlsbad) using Fast SYBR Green Master mix (Applied Biosystems, ThermoFisher). Transcript expression was normalized to the GAPDH house-keeping gene and mRNA[RU, Relative Units] was calculated using the $2^{-\Delta\Delta CT}$ method for qPCR analysis. For comparison of human chemokine expression in A3250 and MDA-MB-231 cell lines, sequences of the PCR primers were as follows: *CCL2*: fp 5′-ATCAC

CAGCAGCAAGTGTC-3′, rp 5′-AGGTGGTCCATGGAATCCTG-3′; CXCL5: fp 5′-GAGAGCTGCGTTGCGTTTG-3′, rp 5′-TTTCCTTGTTTCCACCGTCCA-3′; CCL20: fp 5′-CTGGCTGCTTTGATGTCAGT-3′, rp 5′-CGTGTGAAGCCCACA ATAAA-3′; CXCL2: fp 5′-GGGCAGAAAGCTTGTCTCAA-3′, rp 5′-GCTTCCTCC TTCCTTCTGGT-3′; CXCL14: fp 5′-AAGCCAAAGTACCCGCACTG-3′, rp 5′-GAC CTCGGTACCTGGACACG-3′; CXCL3: fp 5′-AAGTGTGAATGTAAGGTCCCC-3′, rp 5′-GTGCTCCCCTTGTTCAGTATC-3′; CXCL16: fp 5′-GAGCTCACTCGTCCC AATGAA-3′, rp 5′-TCAGGCCCAGCTGCCAGA-3′; CXCL1: fp 5′-GCGCCCAAAC CGAAGTCATA-3′, rp 5′-ATGGGGGATGCAGGATTGAG-3′; GAPDH: fp 5′-CTC TGCTCCTCCTGTTCGAC-3′, rp T5′-TAAAAGCAGCCCTGGTGAC-3′. For comparison of human CCL2 expression in A3250 and five non-IBC cell lines, sequences of the PCR primers were as follows: CCL2: fp 5′-CCCTGTGATGCGGAACTTAT-3′, rp 5′-GATGGCCTTGGTCTTGTTGT-3′.

**mRNA-Seq.** mRNA-Seq #1 (GSE158974) was performed on A3250-control (n = 3) and A3250-shCCL2 (n = 3) tumors in vivo and on A3250 cells in vitro (n = 1). Tumors were harvested when ~4–5 mm in diameter. Samples were preserved in RNAlater™ Solution (Invitrogen AM7020) and processed at the Department of Oncological Sciences Sequencing core for mRNA-Seq (75 bp single-end reads on an Illumina NextSeq 500).

mRNA-Seq #2 (GSE 180788) was performed on A3250, MDA-MB-231 and SUM149 cell lines cultured in vitro in standard culture conditions (n = 3). Briefly, cell pellets from subconfluent cultures of A3250, SUM149, and MDA-MB-231 were collected 24 hr after seeding, RNA was isolated using RNAeasy kit (Qiagen), processed for mRNA-Seq and sequenced at GeneWiz (150 bp paired-end reads on an Illumina HiSeq 2500). The mRNA-Seq datasets generated during the current study are available in the Gene Expression Omnibus database: GSE158974 and GSE180788.

**mRNA-Seq data analyses.** mRNA-Seq fastq files were processed on Galaxy servers[53]. Reads from human and mouse genes were separated by aligning them to the human (hg38) and mouse (mm10) genomes using RNA STAR (v. 2.7.9a)[54]. Gene expression was estimated using featureCounts (Subread v. 2.0.3)[55], which was normalized for gene length to give expression in units of Fragments Per Kilobase of exon per Million mapped reads (FPKM).

Differential gene expression analysis was performed using the edgeR[56] package with a biological coefficient of variation value of 0.1. Only genes with a minimum of 10 reads, absolute log2 fold-change > = 2 in A3250 compared to SUM149 and MDA-MB-231 cell lines and FDR < = 0.05 were considered to be differentially expressed. Venn diagrams were constructed based on the overlapping differentially expressed transcripts.

For the heatmaps and trees, mRNA-Seq data was downloaded from Gene Expression Omnibus (GSE27003)[57] for MDA-MB-468, BT20, HCC3153 and BT474 cells, and GSM3145605 for IBC3. Complex Heatmap library in R was used to generate the trees and heatmaps based on transcript expression levels determined by Kallisto[58].

**IPA pathway analysis.** We performed canonical pathway analysis and upstream regulator analyses on the differentially expressed genes (DEGs) with an absolute log2 fold change > 2 using Ingenuity Pathway Analysis (IPA, Qiagen, Redwood City, www.qiagen.com/ingenuity). We used IPA canonical pathway analysis to determine pathways significantly enriched among DEGs (p value of Fisher's exact test < 0.05; −log p value > 1.3). We considered canonical pathways with an activation z-score ≥ 2 to be activated and pathways with an activation z-score of ≤ −2 to be inhibited[59]. Ratio is calculated as the number of DEGs for the given pathway divided by the total number of genes that map to the same pathway.

We identified significant upstream regulators (p value of Fisher's exact test < 0.05) and determined whether they were significantly activated (activation z-score > 2) or inhibited (activation z-score < −2) using IPA upstream regulator analysis. We excluded the following groups from the analysis results: biologic drug, chemical - endogenous mammalian, chemical - endogenous non-mammalian, chemical—kinase inhibitor, chemical – other, chemical drug, chemical reagent, chemical toxicant, mature microRNA and microRNA.

**Gene set enrichment analysis.** Gene set enrichment analysis was performed using the GSEA software (version 4.1.0)[60,61]. We utilized the GSEA Preranked feature to compare enrichment between the samples. For each comparison, log fold change for each gene was computed using DeSeq2[62], retrieved from https://doi.org/10.1186/s13059-014-0550-8. The genes were then organized into upregulated (log FC > 1) and downregulated (log FC < 1) lists. Each list was sorted by −log(adjusted p value) with the most significantly changed genes at the top of the list, and both lists were input individually into GSEA Preranked and matched against the Hallmark gene set from MSigDB[17]. Statistical significance (nominal p values) of enrichment scores are estimated using an empirical phenotype-based permutation test procedure within the GSEA software[60]. Gene sets were considered significantly enriched with FDR < 25%, and p < 0.05.

**Human IBC transcriptome analyses and statistics.** To obtain all publicly available expression data for inflammatory breast cancer (IBC), we conducted a comprehensive search of the Gene Expression Omnibus (GEO)[63], Short Read Archive[64] and EBI Expression Atlas databases[65], as well as The Cancer Genome Atlas (TCGA; https://cancergenome.nih.gov/) and International Cancer Genome Consortium (ICGC; https://dcc.icgc.org/) data portals. No mRNA-Seq datasets were found. We selected E-MTAB-1006 and GSE22597 microarray datasets (from EBI Array Express and GEO databases, respectively) because they had a large number of IBC samples, because tumor samples were obtained prior to treatment, and because gene expression profile was generated on breast cancer samples in all different hormone receptor categories[24–26,66].

We downloaded the processed expression data for IBC and non-IBC tumor samples and obtained the normalized expression values for all genes, where the raw data was previously normalized using gcrma package in BioConductor (R version 2.13.0)[67]. We assessed the expression of genes represented by multiple probes using the probe with the maximum signal intensity. We then calculated the percentile rank of signal intensity for each of the analysed genes in each of the samples. Percentile ranks of CCL2 were calculated as described above and the significance of the difference in distributions of percentile ranks in IBC vs non-IBC samples was assessed using Wilcoxon rank sum test. To assess macrophage enrichment, we calculated the percentile rank of gcrma normalized signal intensity in IBC and non-IBC tumor samples for 37 TAM-specific genes[30]. For each gene, the percentile ranks were z-score transformed. The samples were hierarchically clustered using Ward's clustering criterion and Euclidean distance. Samples were divided into two clusters and the difference of frequencies of IBC samples in the two clusters compared using chi-square test.

**Statistical analyses.** Statistical differences in proliferation (EdU) and F4/80+ cell densities in tissue sections were determined through an unpaired parametric two-tailed t test. Differences in necrosis were determined using a one-sample two-tailed t test. The statistical difference in weight loss between tumor-naïve and tumor-bearing mice was determined through a one-sample t test. For flow cytometry analysis, significant differences between cell subset fractions were determined using a goodness of fit linear regression model. Kruskal-Wallis non-parametric rank sum test was carried out to determine significance between absolute values of cell subsets. In the case of tumors and lymph nodes, where multiple tissues originated from the same mouse, a linear mixed model fit by REML and t test using Satterthwaite's method were used to control for repeated measures and determine significant differences between groups. Differences between absolute values of cell subsets were determined with a Wilcoxon rank sum test (Mann-Whitney non-parametric test). Tumor bioluminescence data was first log transformed to create symmetry, followed by a linear mixed model fit by REML and t test using Satterthwaite's method to test differences between the groups. Bioluminescence in the organs was first tested for normality, and significant differences were determined by a Mann-Whitney two-tailed t test.

**Reporting summary.** Further information on research design is available in the Nature Research Reporting Summary linked to this article.

## Data availability

The mRNA-Seq datasets generated in this study have been deposited in the Gene Expression Omnibus (GEO) database under accession code GSE158974 and GSE180788. Publicly available mRNA-Seq data used in this study are available in the Gene Expression Omnibus database under accession code GSE27003 for MDA-MB-468, BT20, HCC3153 and BT474 cells and GSM3145605 for IBC3. Microarray datasets for the human IBC transcriptomic analysis conducted in this study are available in the EBI ArrayExpress database under accession code E-MTAB-1006 as well as the GEO database under accession code GSE22597. Remaining data are available within the Article, Supplementary Information, or the Source Data provided with this paper.

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

## Acknowledgements

This research was supported by grants to S.A.A. (P01CA80058, R01CA170702 and UO1 OHO11328) and from the Breast Cancer Research Foundation, and to M.S. from the Susan G. Komen and Milburn Foundations (IBC17512321), Eli Lilly, and the Kathryn Crosby Cancer Research Initiative. The authors also wish to acknowledge support from the Tisch Cancer Institute Shared Resource Facilities (NCI Cancer Center Support Grant P30 CA196521-01) including: Jordy Ochando, Flow Cytometry Core; Jose Javier Bravo-Cordero, Microscopy Core; Jiayi Ji, Biostatistics Facility; Sungyee Kim-Schultz, Human Immune Monitoring Center; and Saboor Hekmaty, Dept. of Oncological Sciences NextSeq Sequencing Facility. We thank Drs. Bronek Pytowski and Dolores Hambard-zumyan for critical reading of the manuscript.

## Author contributions

A.R. and S.D. participated in the experimental design, performed animal experiments, Fluc imaging, histopathology, immunostaining, image acquisition and analyzed data; A.R. also helped write the manuscript; I.P. and L.G. participated in the experimental design, performed cell and molecular biology experiments, and analyzed data; L.G. also generated the expression constructs and transduced cell lines utilized; R.F.R. performed animal experiments, immunostaining, whole-mount tissue imaging, image acquisition, and analyzed data; A.K.E. performed FACS experiments, immunostaining, and analyzed data; A.S., R.K. and R.S. performed bioinformatic analyses of mRNA-Seq data; R.S. developed the software for STR analysis, and. R.K. also performed microarray gene expression analyses; S. Y. and R.Q. performed animal experiments and histology; S.Y also performed image acquisition (slide scanning); G.A. and S.J. evaluated pathology of human IBC specimens; M. S. designed and supervised the study, analyzed data and wrote the manuscript; S.A.A. initiated, designed and supervised the study, revised and edited the manuscript.

## Competing interests

The authors declare no competing interests.
