## [Peer Review File · Nature Communications]

Reviewers' Comments:

Reviewer #1:

Remarks to the Author:

Comments for the Authors

Overview: In this article, Rogic A. et al., described a novel human triple negative inflammatory breast cancer (TN-IBC) cell line, A3250, that recapitulates in a mouse xenograft model the aggressive metastatic phenotype of human IBC in a mouse orthotopic xenograft model. This has long been a goal of breast cancer research. They show that A3250 cells expressed high levels of CCL2, a macrophage recruiting chemokine, to both primary and metastatic tumor sites in IBC, in the absence of detectable levels of its cognate receptor. In addition, tumor associated macrophages were particularly enriched in A3250 tumors and CCL2 knockdown, leading to reduced macrophage densities, tumor cell proliferation and metastasis. In addition, CCL2 was identified as a key factor driving macrophage expansion and promoting A3250 tumor growth. The A3250 human IBC model provides a unique opportunity to uncover key aspects of IBC biology that were not approachable previously. Overall, the study is important and the results are interesting. There are however. A number of revisions that are required, and many areas where text and figures do not agree.

Major comments:

1. After confirming that A3250 cells are IBC and TNBC, the authors should also compare their data with another TN-IBC cell line, such as SUM-149 cells or other human published IBC PDX models, if these data are available.
2. Where are the results shown describing the analysis of tumor cell proliferation by EdU pulse described on from Page 6, lines 10-11?
3. Figure 1B, include a label for the scale bar, what are those numbers representative of (FC?).
4. Magnification index should be included in figures as appropriate.
5. Other IBC cells in mice (i.e. KPL4 cells) and IBC patients have shown to develop cachexia. Was the loss in weight shown in Fig. 2K a result of cachexia development?
6. Figure 1E is not a brightfield image – should be corrected.
7. Figure 1F, is blurry and does not convey what the authors are attempting to describe.
8. Supplemental Figure 1 legend refers to 'HER2 immunostaining of A3250' as panel G, which seems to be mislabeled.
9. Top of page 5 refers to low levels of HER2 mRNA in the A3250 cell line, however in the supplemental material, it is indicated to be a log2 of approximately 5, what is considered to be low, 5 isn't suggestive of low levels.
10. Figure 3A- text doesn't match the manuscript text (legend indicates 4 weeks, manuscript text states 3-4 weeks), please clarify.
11. Top of page 8 indicates Figure 3 C-D are the same timepoint as Figure 3A-B, however in the legend A-B it is indicated that they were taken at 4 weeks, and C-D were taken at 3 weeks. Please clarify.
12. Top of page 10 refers to 'Fig. 3O-R', however Fig. 3P-R does not exist.
13. The A3250 tumors shown in Fig. 3G-I and Suppl. Fig. 2D-F, should also be stained with GFP if possible to distinguish the tumor from the F4/80+ macrophages, and then merge the two images.
14. What staining was used for Suppl. Fig. 2P?
15. What is the difference between Ly6C vs Ly6G? This is not described or mentioned in the manuscript.
16. In the results section (Page 9, line 17), the sentence describing blood levels from Fig. 4J, how was the increase percentage calculated? As seen in Fig. 4J, there may be a 32% increase of total CD11b+ cells, but a 104% and 152% increase for Ly6Chigh and Ly6Ghigh, respectively, is not shown.
17. The Fig. 4I-J are being compared to the Naïve tumor, shouldn't they be compared with the MDA-MB-231 model?
18. Figure 5E suggests A3250 CCL2 expression is compared to 'five non-IBC cell lines', however only three are shown. They should all be identified and shown.
19. Figure 5F needs to be adjusted as A3250 is within the plotting area.
20. Figure 5H – it is hard to read lower left quadrant numbers.
21. In Figure 5J, multiple bands are observed. Please indicate which is CCL2 or why there are multiple bands observed.

Minor comments:

1. In Suppl. Table 1, what do columns B and C describe? Is column B showing all human cell lines and column C the A3250 cells?
2. What is the scale bar for Suppl. Fig. 1A-C? This is not described in Figure legends for Suppl. Figures.
3. Additional biological replicates should be added for in vitro data from Suppl. Table 2, 6, 7 and 8.
4. The Suppl. Fig. 1H and 1I are incorrectly labeled in the Suppl. Figure Legend description.
5. What number of cells were orthotopically injected for studies shown in Fig. 1C, D? The results section (page 5, line 13) says 10^5 , but Figure legend says 10^6 .
6. The results section (Page 5, lines 13-14) describes the incidence of primary tumor formation for either SCID or NOD-SCID mice. However, Fig. 1C and 1D only show SCID mice incidence. Could Fig. 1C and 1D, also include the NOD-SCID mice incidence data?
7. How many biological replicates were used for A3250 tumor cells in vitro for the RNAseq analysis shown in Suppl. Table 3 and Suppl. Table 4? This is not mentioned.
8. The Fig. 2A shows 17/35 LNs positive, not 19/35 as mentioned in the results section (Page 7, line 4). This should be corrected.
9. The Suppl. Fig. 2J-L are incorrectly labeled in Suppl. Figure legend section. This should be corrected.
10. Which mice (SCID or NOD-SCID) were used for experiments shown in Fig. 3 and Fig. 5L-N? This is not mentioned.
11. In the results section (Page 8, lines 20-23), this sentence could include the percentage of cells shown in Fig. 3J or an average of percentage from Fig. 3L. Also, Fig. 3O should include the percentage in numbers, since the y-axis is not described.
12. According to Suppl. Table 4, CCL24 was slightly detected in the A3250 model, contrary to that mentioned in the manuscript (Page 11, lines 4-6), which says it was undetected.
13. The sentence describing the expression of CXCL5 in A3250 cell in vitro compared to the eight non-IBC cell lines (Page 11, lines 12-14), should address Suppl. Table 5, not Suppl. Table 4. This should be corrected.
14. In page 12 lines 1-4, the manuscript describes the number of cells expressing CD11b+, Ly6C high and Ly6C low in the blood, how was the percentage of Ly6Chi (80%) and Ly6Clo (18%) monocytes determined?
15. Why is the blood data from Suppl. Fig. 3B only show CD11B+ and Ly6C-F4/80-, and not F4/80+Ly6Clo and Ly6Chi, like the tumors (Suppl. Fig. 3C) and lungs (Suppl. Fig. 3D) data?
16. The Fig. 5E shows three non-IBC cell lines, not five as mentioned in Figure legend section (Page 44). This should be corrected.
17. In page 13 lines 10-12, the manuscript describes tumor proliferation reduction by CCL2 KD. How was the reduction of 83% determined?
18. The results section describing Fig. 7 immunostaining shows three human IBC, not nine as mentioned in the article (Page 16, line 8). This should be corrected.

Reviewer #2:

Remarks to the Author:

In this study, the authors established a cell line, named A3250, from triple-negative IBC patients. This cell line formed a metastatic tumor in SCID mice with recapitulating IBC features, such as erythema, nipple changes, invasion into the subcutaneous fat, and dermal lymphatic vessel invasion with lymphovascular emboli. Using this mouse model, authors demonstrated the recruitment of inflammatory cells into the tumor and adjacent skin early in tumor development, before dermal vessel invasion. CD11b+ F4/80+ macrophages and CD11b+Ly6ChiLy6G-F4/80- monocytes, known as mMDSCs, were prominent immune cells A3250 tumor, whereas CD11b+Ly6G+Ly6CloF4/80- TANS were the most highly represented subset of myeloid cells in the MDA-MB-231 tumor.

The authors found that the chemokine gene expression profile of A3250 was similar to those of IBC patients. The mRNA level of CCL2, a well-known chemoattractant for macrophages and monocytes, was the highest expressed chemokine in A3250 tumor cells. The knockdown of CCL2 in A3250 reduced tumor cell growth and lung metastasis and macrophages' infiltration into the tumor

and infiltration of Ly6C+ inflammatory monocytes into the lungs. Based on the bioinformatical analysis of IBC patient gene expression, the authors claim that human IBCs are macrophage enriched.

The xenograft model using A3250 well recapitulates IBC features. This newly established IBC cell line may be a useful tool for IBC research. However, such cell lines were already reported in previous reports, such as MARY-X. Therefore, the novelty of this A3250 is limited.

Much of what is described in this study is previously reported and is not unique to IBC. Thus, the impact of this study is very small. Authors should consider why A3250 represents the IBC-like phenotype by comparing to non-IBC.

The following concerns are some of the major issues of this paper.

- Fig. 1B: To confirm A3250 is an IBC cell line, they compared it with many non-IBC cell lines. Here only SUM149 is included for comparison, which doesn't make sense. And the pattern is sort of different from SUM149. It lacks comparison with other IBC cells.
- Fig. 1N: inflammatory pathway, JAK/STAT, angiogenesis not enriched in A3250.
- Its triple-negative status should be evaluated by IHC staining or another standard method.
- Analysis of gene expression profiles of IBC patients was not compared with non-IBC patients. Therefore, it is not clear whether the result is IBC-specific or not, such as macrophage-rich gene expression and high CCL2 expression.
- In this study, only mRNA expression was examined for chemokines. Their expression at the protein level should be assessed by ELISA or other methods.
- The reason why MDA-MB-231 was used as control is unclear. Is MDA-MB-231 neutrophil-enrich?
- How does the tumor begin to look like when their CCL2 were knocked out? Did they lose the IBC features such as erythema or tumor emboli?
- All experiments should be done with at least one more different IBC cell.

RESPONSE TO REFEREES

We thank the reviewers for their constructive comments, and we have revised the manuscript in accordance with their suggestions including the addition of a number of new results. Our responses to the Reviewers' comments are indicated below:

REVIEWER COMMENTS

Reviewer #1: *This has long been a goal of breast cancer research. They show that A3250 cells expressed high levels of CCL2, a macrophage recruiting chemokine, to both primary and metastatic tumor sites in IBC, in the absence of detectable levels of its cognate receptor. In addition, tumor associated macrophages were particularly enriched in A3250 tumors and CCL2 knockdown, leading to reduced macrophage densities, tumor cell proliferation and metastasis. In addition, CCL2 was identified as a key factor driving macrophage expansion and promoting A3250 tumor growth. The A3250 human IBC model provides a unique opportunity to uncover key aspects of IBC biology that were not approachable previously. Overall, the study is important and the results are interesting. There are however a number of revisions that are required, and many areas where text and figures do not agree.*

We appreciate Reviewer#1's very positive comments and their careful review has helped us to strengthen the manuscript with additional findings. Our responses are as follows:

Major comments

1. *After confirming that A3250 cells are IBC and TNBC, the authors should also compare their data with another TN-IBC cell line, such as SUM-149 cells or other human published IBC PDX models, if these data are available.*

Agree. We have revised the manuscript to include mRNA-Seq analyses of A3250 with SUM149 and IBC-3 by hierarchical clustering and more detailed comparison of differentially expressed genes, IPA canonical pathways, upstream regulators, and Gene Set Enrichment Analyses (GSEA) for select pathways of interest of A3250 and SUM149 (new Fig. 3, new Suppl. Tables 5-8).

At the biological level, we have also revised the manuscript to include comparative analyses of A3250 and SUM149 tumor characteristics and inflammatory phenotype. Tumors were evaluated for inflammation, including macrophage infiltration of primary xenograft tumors (revised Fig. 4, revised Suppl. Fig. 4), erythema (new Suppl. Fig. 6), gross tumor appearance (revised Suppl. Fig. 2), pattern of invasive growth, and dermal lymphatic vessel invasion (revised Suppl. Fig. 2). For each parameter, A3250 shows a more florid, aggressive phenotype than SUM149.

We have also compared A3250 chemokine expression to SUM149 and IBC3 by mRNA-Seq (revised Suppl. Table 10), and to SUM149 by qPCR (revised Fig. 7B, C). Chemokine protein expression examined by Olink proteomics, revealed that CCL2 protein is expressed and secreted at high level by A3250 but not SUM149 (revised Fig. 7, new Suppl. Table 11).

2. *Where are the results shown describing the analysis of tumor cell proliferation by EdU pulse described on from Page 6, lines 10-11?*

We have added new Suppl. Fig. 3 to show representative images of A3250 tumors used for EdU quantification, and we have added a table with the raw data (new Suppl. Table 3). The results now indicate “Analysis of tumor cell proliferation by EdU pulse independently revealed a high proliferation rate, with an average of 458 Edu+ tumor cells per mm² tumor/ 6hr (STD +/- 70.2) (page 7). Corresponding methods are described on page 26.

3. *Figure 1B, include a label for the scale bar, what are those numbers representative of (FC?).*

Agree. In the revised manuscript, Fig. 1B is replaced with new data in response to the Reviewer’s question #1. A new heatmap is shown in Fig. 3A. The numbers on the label represent z-score as indicated in the legend to Fig. 3A.

4. *Magnification index should be included in figures as appropriate.*

We included scale bars on each image to provide information on magnification. Because images were acquired with several different microscopes, cameras and using the slide scanner, there would be a range of magnification indexes to show. For clarity, we are showing scale bars at standard sizes, mostly at 100 μ m.

5. *Other IBC cells in mice (i.e. KPL4 cells) and IBC patients have shown to develop cachexia. Was the loss in weight shown in Fig. 2K a result of cachexia development?*

We consistently observed weight loss, wasting of skeletal muscle and inflammation in A3250 tumor-bearing mice, which are indicators of cachexia according to the Animal Cachexia Score (Betancourt A. et al. The animal cachexia score ACASCO). *Animal Model Exp Med.* 2019;2(3):201-209.). We have revised the text on page 8 in the Results section as follows: “Associated with the high metastatic burden at endpoint, we observed notable weight-loss (Fig. 2K) and muscle-wasting (not shown) of tumor-bearing animals, consistent with cachexia”.

6. *Figure 1E is not a brightfield image, it should be corrected.*

Agree. This image was taken with a standard camera. We have revised the figure legend and referred to the image in Fig. 1E as a “representative picture”.

7. *Figure 1F, is blurry and does not convey what the authors are attempting to describe.*

We have replaced the original image with a new one, which we hope better conveys the overall appearance of the A3250 IBC tumor as flat (pancake-like) with an irregular shape. Nonetheless, images were taken through skin, whose thickness makes it difficult to correct for optical distortions, resulting in a somewhat blurred image, particularly at the edges.

8. *Supplemental Figure 1 legend refers to HER2 immunostaining of A3250 as panel G, which seems to be mislabeled.*

Agree. Suppl. Fig. 1 has been revised, and HER2 staining is now shown in panels G and J. The legend has been revised accordingly.

9. *Top of page 5 refers to low levels of HER2 mRNA in the A3250 cell line, however in the supplemental*

material, it is indicated to be a log₂ of approximately 5, what is considered to be low, 5 isn't suggestive of low levels.

While we indicated in the text that A3250 cells expressed low levels of HER2 mRNA, comparable to other TNBC cell lines, we agree with the reviewer that mRNA level of 28 fpkm is not a low level per se. However, it is much lower than the mRNA expression level in cell lines considered HER2+. For example, BT474 with amplified HER2 shows 77-fold higher expression than A3250 (1939 fpkm vs. 28 fpkm, respectively). In an effort to clarify this point, we have revised the text as follows:

“They lacked detectable estrogen or progesterone receptors (ER or PR), and expressed HER2 (ERBB2) mRNA at levels comparable to other TNBC cell lines (Suppl. Fig. 1D). In comparison, BT474 cells with amplified HER2, expressed more than 70-fold higher levels (Suppl. Fig. 1D, Suppl. Table 2).” (Results, page 5).

0. *Figure 3A- text doesn't match the manuscript text (legend indicates 4 weeks, manuscript text states 3-4 weeks), please clarify.*

Agree. Old Fig. 3A (now new Fig. 4A), shows tumors at 4 weeks, as indicated in the legend and in the corrected text on page 10:

“Immunostaining demonstrated that A3250 tumors were heavily infiltrated with CD45+ cells four weeks post-injection, in comparison to MDA-MB-231 tumors (Fig. 4A and B).”

1. *Top of page 8 indicates Figure 3 C-D are the same timepoint as Figure 3A-B, however in the legend A-B it is indicated that they were taken at 4 weeks, and C-D were taken at 3 weeks. Please clarify.*

The legend correctly indicates the timepoints (please note that old Fig. 3 is new Fig. 4): Fig. 4A, B at 4 weeks, and Fig. 4C, D at 3 weeks. We have modified the text in an effort to increase clarity as follows: “In the dermis, CD45+ cells were already abundantly present at three weeks post-injection (Fig. 4C). Dermal lymphatics were numerous and enlarged at that same time-point, but did not yet contain tumor emboli (Fig. 4D). (Results, pages 10, 11)

2. *Top of page 10 refers to Fig. 3O-R, however Fig. 3P-R does not exist.*

Thanks for pointing out this omission which we have now corrected. We have revised the text (now on page 12) as follows: “In contrast, in the MDA-MB-231 tumor model Ly6G+ neutrophils were the main cell subset recruited to both, primary tumors (Fig. 5C, E, F) and to the lungs (Fig. 6K).”

Please note that in the revised manuscript the Figure numbers have changed.

3. *The A3250 tumors shown in Fig. 3G-I and Suppl. Fig. 2D-F, should also be stained with GFP if possible to distinguish the tumor from the F4/80+ macrophages, and then merge the two images.*

Agree. We stained tumors with pan-cytokeratin antibody to visualize tumor cells because A3250 tumors in this experiment were from the parental cells. Double immunostaining for cytokeratin and for F4/80 in A3250 tumors is now shown in revised Fig. 4G, H. Note that tumor-adjacent skin in the Fig. 4I was not stained for cytokeratin, because there is no tumor present in this skin sample. In the Supplement, we have included DAPI with the

images of tumors stained for F4/80 (revised Suppl. Fig. 4D-L). Please note that the Figure numbers have changed in the revised manuscript.

14. *What staining was used for Suppl. Fig. 2P?*

DAPI nuclear staining was used in the old Suppl. Fig. 2P. However, we have removed that image in the revised manuscript because DAPI stain is already shown together with the immunostaining for Ly6G in the same sample (revised Suppl. Fig. 4O).

15. *What is the difference between Ly6C vs Ly6G? This is not described or mentioned in the manuscript.*

Lymphocyte antigen-6 (Ly6) proteins Ly6C and Ly6G belong to a superfamily of transmembrane proteins and were used in combination with other markers to define the subsets of myeloid lineage. Expression of Ly6G is restricted to murine neutrophils, whereas monocytes express Ly6C, but not Ly6G. The level of Ly6C also distinguishes between the monocyte subsets: Ly6C^{hi} monocytes (i.e. immature or inflammatory monocytes) migrate from the bone marrow to the site of inflammation, whereas Ly6C^{lo} monocytes (i.e. mature monocytes) are precursors for resident macrophages in tissues.

Because of space constraints, we added the references on Ly6C and Ly6G and their use as surface markers to the relevant sentence in the Results section (page 11): “Flow cytometry demonstrated that A3250 tumors contained two major populations of CD11b+ myeloid cells at four weeks post-injection, based on Ly6C and Ly6G in combination with other markers to define the myeloid lineage subsets²⁰⁻²².”

20. Kumar, V. *et al.* The Nature of Myeloid-Derived Suppressor Cells in the Tumor Microenvironment. *Trends in immunology* **37**, 208-220, doi:10.1016/j.it.2016.01.004 (2016).

21. Kim, I. S. *et al.* Immuno-subtyping of breast cancer reveals distinct myeloid cell profiles and immunotherapy resistance mechanisms. *Nature Cell Biology* **21**, 1113-1126, doi:10.1038/s41556-019-0373-7 (2019).

22. Upadhyay, G. Emerging Role of Lymphocyte Antigen-6 Family of Genes in Cancer and Immune Cells. *Front Immunol* **10**, 819, doi:10.3389/fimmu.2019.00819 (2019).

16. *In the results section (Page 9, line 17), the sentence describing blood levels from Fig. 4J, how was the increase percentage calculated? As seen in Fig. 4J, there may be a 32% increase of total CD11b+ cells, but a 104% and 152% increase for Ly6Chigh and Ly6Ghigh, respectively, is not shown.*

The increase in percentage of each immune cell subtype was calculated as follows: (final value – initial value) / initial value, where final value represents A3250 tumor-bearing mouse and initial value represents naïve mouse control. Mean values were used in calculations:

Blood CD11b+/CD45+	(89.58-67.52)/67.52	= 32.7%
Blood Ly6C ^{hi} _{g^h} /CD11b+	(11.85-5.82)/5.82	= 104%
Blood Ly6G ^{hi} _{g^h} /CD11b+	(53.28-21.14)/21.14	= 152%

These findings show increase in fraction of CD11b+, Ly6C^{hi}_{g^h}, and Ly6G^{high} cells in blood of A3250 tumor-bearing mice. Please note that old Fig. 4J is now revised Fig. 6J.

17. *The Fig. 4I-J are being compared to the Naive tumor, shouldn't they be compared with the MDA-MB-231 model?*

In the lung, inflammatory cells were associated with metastatic foci, so that flow cytometry data from the entire lung could not be compared between mice with different metastatic burden. For this reason, we did not directly compare A3250 lung flow cytometry data to MDA-MB-231. However, we did examine inflammatory infiltrates associated with metastases of A3250 and MDA-MB-231 tumors by immunostaining (old Fig. 4A-H is now revised Fig. 6A-H).

We also included data showing frequencies of Ly6C+ and Ly6G+ cells in blood from MDA-MB-231 tumor-bearing mice in Fig. 6J. Please note that old Fig. 4 is now revised Fig. 6.

18. *Figure 5E suggests A3250 CCL2 expression is compared to five non-IBC cell lines, however only three are shown. They should all be identified and shown.*

In response to the reviewer's question, we have now compared A3250 CCL2 expression by qPCR to nine cell lines including SUM149 (IBC) and eight non-IBC cell lines (see new Fig.7C). We also show mRNA-Seq data for CCL2 expression in A3250 compared to SUM149, IBC3, MDA-MB-231, HCC3153 and BT-20 cells in revised Suppl. Table 10.

Corresponding results are described in the revised manuscript (page 13) as follows: "CCL2 mRNA was also highly upregulated in A3250 cells not only in comparison to SUM149, but also to IBC3 and to a panel of non-IBC breast cancer cell lines (Fig. 7C, Suppl. Table 10)".

19. *Figure 5F needs to be adjusted as A3250 is within the plotting area.*

Agree. This Figure has been revised to remove the misplaced A3250 label. Please note that old Fig. 5F is now new Fig. 7H.

20. *Figure 5H it is hard to read lower left quadrant numbers.*

Agree. In response, we have increased the size of the lower quadrant numbers and adjusted their position in order to make them easier to read. Please note that old Fig. 5H is new Fig. 7J.

21. *In Figure 5J, multiple bands are observed. Please indicate which is CCL2 or why there are multiple bands observed.*

The Western blot shows three bands, at 11 kDa, 13 kDa and 15 kDa. According to the UniProt protein database, human CCL2 is comprised of 99 amino acids, of which 23 are a signal peptide. CCL2 has a predicted molecular weight of 11 kDa according to Ensembl database (<https://bit.ly/2UTssGU>). However, it can be glycosylated on Asp37 rendering proteins of several higher molecular weights (13, 15, 15.5, 16, and 18 kDa) depending on the level of glycosylation (*Rollins BJ, Stier P, Ernst T, Wong GG. The human homolog of the JE gene encodes a monocyte secretory protein. Mol Cell Biol. 1989; 9(11):4687-95.*). Therefore, in our Western the 11kDa band is the native full-length form, whereas the 13 kDa and the 15 kDa bands are most likely glycosylated forms. We have revised the text to clarify these points as follows:

“CCL2 protein showed multiple bands on Western; including a 11kDa native form and 13 kDa and 15 kDa glycosylated forms” (Results, page 15, reference #29). Please note that old Fig. 5J is new Fig. 8B.

Minor comments

1. *In Suppl. Table 1, what do columns B and C describe? Is column B showing all human cell lines and column C the A3250 cells?*

Suppl. Table 1 shows results of Short Tandem Repeat (STR) DNA profiling of A3250 cells. Values shown in columns B and C represent allele scores. Allele score is the number of tandem repeats (STR) present in each allele of the given genetic locus. Column B shows number of tandem repeats for allele 1, and column C shows number of tandem repeats for allele 2 for the given locus. For example, genetic locus D3S1358 on chromosome 3 has an allele score of 15, 18. This means that there are 15 tandem repeats in allele 1 and 18 tandem repeats in allele 2 for this locus. One allele score is shown in cases where only one allele was present, for example for locus D5S818 in this sample.

Both ATCC's and the Mount Sinai DNA sequence Core's cell line authentication is based on 17 STR markers plus amelogenin for gender determination, which creates a unique genetic signature. The STR profile of A3250 shown in Table 1 does not match any of the cell lines in the ATCC reference database. We have revised the legend to this table in an effort to clarify this point.

2. *What is the scale bar for Suppl. Fig. 1A-C? This is not described in Figure legends for Suppl. Figures.*

The scale bar is 100 μ m, which is now described in the revised legend. Please note that Suppl. Fig. 1C which provides salient findings is now Fig. 1B in the revised Figures. Suppl. Fig. 1A and B have been removed because they provided essentially the same information already presented in Fig. 1B.

3. *Additional biological replicates should be added for in vitro data from Suppl. Table 2, 6, 7 and 8.*

Agree. We have performed new mRNA-Seq for A3250, SUM149 and MDA-MB-231 cells with n=3, and these data are provided in revised Suppl. Tables 2, 9, 10, and 12.

4. *The Suppl. Fig. 1H and 1I are incorrectly labeled in the Suppl. Figure Legend description.*

Agree. Corrections made. Please note that Suppl. Fig. 1 has been revised, and panels H and I have been replaced with the new pictures of HER2 staining (revised Suppl. Fig. 1G, J). Figure legend has been revised accordingly.

5. *What number of cells were orthotopically injected for studies shown in Fig. 1C, D? The results section (page 5, line 13) says 10^5 , but Figure legend says 10^6 .*

Agree. Corrections made as follows in text (page 6):

“Orthotopic injection of 10^6 cells into the mammary fat pad of either SCID or NOD-SCID mice led to the formation of primary tumors at 100% incidence, with comparable growth kinetics (Fig. 1C, D).”

6. The results section (Page 5, lines 13-14) describes the incidence of primary tumor formation for either SCID or NOD-SCID mice. However, Fig. 1C and 1D only show SCID mice incidence. Could Fig. 1C and 1D, also include the NOD-SCID mice incidence data?

Agree. We have revised Fig. 1C to include A3250 tumor growth curves in both SCID and NOD-SCID mice. Results section has been revised to try to increase clarity as follows (page 6): “Orthotopic injection of 10^6 cells into the mammary fat pad of either SCID or NOD-SCID mice led to the formation of primary tumors at 100% incidence, with comparable growth kinetics (Fig. 1C, D).”

Figure 1 legend has been revised as follows:

(C) Tumor growth following orthotopic injection of 10^6 A3250 tumor cells in SCID or NOD-SCID mice measured by caliper weekly. Data represent mean + SEM. (D) Bioluminescence imaging of A3250 tumors in SCID mice at week six (n=12 tumors). Black line denotes mean value.

7. How many biological replicates were used for A3250 tumor cells in vitro for the RNAseq analysis shown in Suppl. Table 3 and Suppl. Table 4? This is not mentioned.

In response, we have revised the manuscript to indicate the number of replicates in the tables. In old Suppl. Table 3 (now Suppl. Table 4), pathway analysis for tumors was done based on three replicates, and for cells in vitro based on one replicate from the same mRNA-Seq experiment. We also performed new mRNA-Seq experiments on cells in vitro with three replicates, and we have included the data from both experiments in revised Suppl. Table 9 (old Suppl. Table 4).

8. The Fig. 2A shows 17/35 LNs positive, not 19/35 as mentioned in the results section (Page 7, line 4). This should be corrected.

Correction made on revised page 7 as follows: “A3250 cells metastasized to the sentinel (inguinal) lymph nodes (LNs) at 100% incidence and to more distant LNs (axillary and brachial) at 46% incidence within six weeks (16/35 LNs positive) (Fig. 2A).”

Please note that 16 LNs (not 17 LN) were positive based on criteria of Fluc > 10^4 . We analyzed two lymph nodes (axillary and brachial) on the left (L) and right (R) side of each mouse. The table with the raw data is presented below, LNs positive for metastases are indicated in red:

Mouse#	1	2	3	4	5	6	7	8	9
L LN1	3.60x10 ²	9.83x10⁶	9.25x10 ³	6.88x10⁵	3.28x10 ³	3.13x10 ³	5.04x10 ³	7.76x10 ²	1.15x10⁶
L LN2	1.56x10 ³	4.48x10⁵	2.51x10⁵	9.94x10 ³	2.14x10 ³	3.65x10 ³	8.90x10 ³	3.58x10⁴	7.69x10 ³
R LN1	N/A	1.38x10⁶	1.86x10⁵	5.42x10⁴	5.76x10⁴	4.51x10 ³	1.40x10⁴	1.17x10 ³	2.86x10⁶
R LN2	6.27x10⁴	1.08x10⁵	9.96x10⁶	8.01x10 ³	5.26x10 ³	3.61x10 ³	2.92x10 ³	9.90x10 ¹	1.56x10⁵

9. The Suppl. Fig. 2J-L are incorrectly labeled in Suppl. Figure legend section. This should be corrected.

Agree. However, because of the space constraints, we have removed Suppl. Fig. 2J-L showing macrophage proliferation and the related text.

22. Which mice (SCID or NOD-SCID) were used for experiments shown in Fig. 3 and Fig. 5L-N? This is not mentioned.

Agree. Correction made. We have added the following sentence to the Methods (page 23): “SCID mice were used in all experiments unless indicated otherwise.” NOD SCID mice were used only in one experiment to compare A3250 tumor growth to that in SCID mice (Fig. 1C). Please note that data in old Fig. 3 is now shown in revised Fig. 4 and 5. Old Fig. 5 is revised Fig. 7.

11. In the results section (Page 8, lines 20-23), this sentence could include the percentage of cells shown in Fig. 3J or an average of percentage from Fig. 3L. Also, Fig. 3O should include the percentage in numbers, since the y-axis is not described.

Agree. Average values of percentages shown in old Fig. 3L (now Fig. 5C) are now included in the results section (page 11, 12):

“Tumor-associated macrophages were most abundant (avg 32.2%, median 31.7%), followed by Ly6C^{hi} high inflammatory monocytes (avg 19.5%, median 19.6%). In contrast, tumor-associated neutrophils were the least represented CD11b+ subset (avg 17.6%, median 9.6%).”

Old Fig. 3O (now Fig. 5F) shows the relative ratios of the indicated subsets compared to each other, rather than the percentage of CD11b+ for each subset. Revised Fig. 3O (now Fig. 5F) now includes a y-axis label, and the corresponding legend has been revised in an effort to clarify this point: “Relative ratios of different myeloid cell populations in tumors as indicated.”

12. According to Suppl. Table 4, CCL24 was slightly detected in the A3250 model, contrary to that mentioned in the manuscript (Page 11, lines 4-6), which says it was undetected.

Agree. Sentence deleted and legend revised to clarify the threshold FPKM value to consider the gene expressed(1,2). CCL24 was detected at very low levels, with FPKM values in the range of 0.02-0.08 in our samples (old Suppl. Table 4 is new Suppl. Table 9). We classified this gene as not expressed, because we set the threshold at FPKM > 1 to consider a gene expressed, based on common practice in the literature. The Sequence Quality Consortium assessed the consistencies and relative cutoffs, and reported that FPKM < 1 were not consistently validated by RT-PCR (1, 2).

References:

1. SEQC/MAQC-III Consortium., Su, Z., Łabaj, P. *et al.* A comprehensive assessment of RNA-seq accuracy, reproducibility and information content by the Sequencing Quality Control Consortium. *Nat Biotechnol* **32**, 903–914 (2014).

2. Xu J, Gong B, Wu L, Thakkar S, Hong H, Tong W. Comprehensive Assessments of RNA-seq by the SEQC Consortium: FDA-Led Efforts Advance Precision Medicine. *Pharmaceutics*. 2016 Mar 15;8(1):8.

13. The sentence describing the expression of CXCL5 in A3250 cell in vitro compared to the eight non-IBC cell lines (Page 11, lines 12-14), should address Suppl. Table 5, not Suppl. Table 4. This should be corrected.

Agree. However, the revised manuscript deletes the statement referring to CXCL5 in order to comply with text length restrictions.

14. In page 12 lines 1-4, the manuscript describes the number of cells expressing CD11b+, Ly6C high and Ly6C low in the blood, how was the percentage of Ly6Chi (80%) and Ly6Clo (18%) monocytes determined?

The numbers indicate percentages of Ly6C^{hi} or Ly6C^{lo} monocytes positive for CCR2, not the percentages of Ly6C^{hi} and Ly6C^{lo} monocytes in the blood. In an effort to clarify this point, we have revised the text (page 14) as follows:

“In the blood, CCR2 was expressed by ~30% of total CD11b+ cells, and mainly by Ly6C^{hi} inflammatory monocytes (76% CCR2+ of Ly6C^{hi} on average), suggesting increased mobilization of these cells from the bone marrow. A small fraction of Ly6C^{lo} monocytes (7% CCR2+ of Ly6C^{lo} on average) also expressed CCR2, whereas CD11b+ cells that were Ly6C- were also negative for CCR2”.

We also added representative FACS plots for gating of CCR2+ Ly6C^{hi} and CCR2+ Ly6C^{lo} monocytes in the blood (new Suppl. Fig. 5B). Please note that the numbers on the plots are for the representative sample shown (86%, 18%), whereas the numbers in the text indicate average values.

15. Why is the blood data from Suppl. Fig. 3B only show CD11B+ and Ly6C-F4/80-, and not F4/80+Ly6Clo and Ly6Chi, like the tumors (Suppl. Fig. 3C) and lungs (Suppl. Fig. 3D) data?

Old Suppl. Fig. 3B (revised Suppl. Fig. 5B) shows FACS analysis of CCR2 expression on CD11b+ cell subsets in the blood. We have added two more histograms as requested, to show CCR2 expression on relevant monocyte populations in the blood: Ly6C^{hi} inflammatory monocytes (F4/80^{low} or negative) and Ly6C^{lo} mature monocytes. It can be seen that CCR2 is expressed mainly by Ly6C^{hi} inflammatory monocytes in the blood. We did not include F4/80+/Ly6C^{lo} in the blood dataset because this subset is only found in tissues and not in the blood circulation (Lee, YS. et al. CX₃CR1 differentiates F4/80^{low} monocytes into pro-inflammatory F4/80^{high} macrophages in the liver. *Sci Rep* **8**, 15076, 2018).

16. The Fig. 5E shows three non-IBC cell lines, not five as mentioned in Figure legend section (Page 44). This should be corrected.

Agree. We have also performed additional experiments comparing CCL2 expression by qPCR of A3250 to SUM149 and to eight non-IBC cell lines. These new results are shown in the revised Fig. 7C (old Fig. 5E). The corresponding legend has been **revised as follows**: “qPCR for CCL2 expression in A3250 cells in vitro compared to the SUM149 and eight non-IBC cell lines. **RU = relative units.**”

17. In page 13 lines 10-12, the manuscript describes tumor proliferation reduction by CCL2 KD. How was the reduction of 83% determined?

The percent reduction was calculated as follows: (initial value – final value) / initial value, where initial value represents A3250 control tumor and final value represents A3250 CCL2 KD tumor. Average values of EdU+ cells/mm² tumor were used in calculations.

% reduction in EdU+ cells: (458-78)/458 = 0.829 = 83%

We have added these measurements to new Suppl. Table 3. We also revised the Results to indicate the number of EdU+ cells instead of percent EdU+ area to improve clarity: “CCL2 KD also reduced tumor proliferation by as much as 83%, with an average number of EdU+ cells/mm² tumor decreased from 458 to 78”

per 6 hr" (page 16). Corresponding revised Fig. 8N now indicates the number of EdU+ cells instead of percent area.

18. The results section describing Fig. 7 immunostaining shows three human IBC, not nine as mentioned in the article (Page 16, line 8). This should be corrected.

We performed immunostaining of nine human IBC samples, and our conclusions were based on those results. In old Fig. 7 (revised Fig. 10), we show three samples, each representative of a characteristic IBC histopathology: type 1 – solid tumor, type 2 – emboli in dermal lymphatics, type 3 – diffuse growth. In addition to the three samples shown in Fig. 10, we have now included images of six additional samples stained for macrophages and neutrophils in new Suppl. Fig. 8.

Reviewer #2 (Remarks to the Author): *In this study, the authors established a cell line, named A3250, from triple-negative IBC patients. This cell line formed a metastatic tumor in SCID mice with recapitulating IBC features, such as erythema, nipple changes, invasion into the subcutaneous fat, and dermal lymphatic vessel invasion with lympho -vascular emboli. Using this mouse model, authors demonstrated the recruitment of inflammatory cells into the tumor and adjacent skin early in tumor development, before dermal vessel invasion. CD11b+ F4/80+ macrophages and CD11b+Ly6ChiLy6G-F4/80- monocytes, known as mMDSCs, were prominent immune cells A3250 tumor, whereas CD11b+Ly6G+Ly6CloF4/80- TANs were the most highly represented subset of myeloid cells in the MDA-MB-231 tumor.*

The authors found that the chemokine gene expression profile of A3250 was similar to those of IBC patients. The mRNA level of CCL2, a well-known chemoattractant for macrophages and monocytes, was the highest expressed chemokine in A3250 tumor cells. The knockdown of CCL2 in A3250 reduced tumor cell growth and lung metastasis and macrophages' infiltration into the tumor and infiltration of Ly6C+ inflammatory monocytes into the lungs. Based on the bioinformatical analysis of IBC patient gene expression, the authors claim that human IBCs are macrophage enriched.

The xenograft model using A3250 well recapitulates IBC features. This newly established IBC cell line may be a useful tool for IBC research. However, such cell lines were already reported in previous reports, such as MARY-X. Therefore, the novelty of this A3250 is limited. Much of what is described in this study is previously reported and is not unique to IBC. Thus, the impact of this study is very small. Authors should consider why A3250 represents the IBC-like phenotype by comparing to non-IBC. The following concerns are some of the major issues of this paper.

We thank Reviewer #2 for their comments. In response, we have generated additional results, which support the important role of CCL2 production by A3250 cells in the highly aggressive IBC inflammatory phenotype. The initial manuscript showed that the high level of CCL2 production by these cells, was associated with florid tumor infiltration by macrophages, and the highly aggressive IBC phenotype in vivo and that these features were antagonized by CCL2 knockdown. In response to Reviewers #1 and #2, we show in the revised manuscript that these IBC features were not observed with SUM149 IBC cells, which lack CCL2 production. Finally, we show that clinical IBC samples exhibit high relative CCL2 expression and a macrophage enriched phenotype both by immunostaining and a macrophage RNA signature. All of these new results strengthen our conclusions that A3250 provides a unique human model recapitulating all key features of human IBC and that CCL2 is an important driver of this florid phenotype.

MAJOR COMMENTS

1. *Fig. 1B: To confirm A3250 is an IBC cell line, they compared it with many non-IBC cell lines. Here only SUM149 is included for comparison, which doesn't make sense. And the pattern is sort of different from SUM149. It lacks comparison with other IBC cells.*

Agree. The revised manuscript now includes comparative mRNA-Seq analyses of A3250 with SUM149 and IBC3, two other human IBC lines, by hierarchical clustering (new Fig.3), and a more detailed comparison with SUM149 and MDA-MB-231 of differentially expressed genes, IPA canonical pathways, upstream regulators, and Gene Set Enrichment Analyses (GSEA) for select pathways of interest (new Fig. 3, new Suppl. Tables 58). Data shows that A3250 is most closely related to SUM149 IBC cells.

We have also revised the manuscript to include comparison of A3250 and SUM149 tumor characteristics. Tumors were evaluated for macrophage infiltration of primary tumors (revised Fig. 4, revised Suppl. Fig. 4), erythema (new Suppl. Fig. 6), gross tumor appearance (revised Suppl. Fig. 2), pattern of invasive growth, and dermal lymphatic vessel invasion (revised Suppl. Fig. 2). For each parameter, A3250 shows a more florid, aggressive phenotype than SUM149.

Finally, we have compared A3250 chemokine expression to SUM149 and IBC3 by mRNA-Seq (Suppl. Table 10), and to SUM149 by qPCR (revised Fig. 7B, C). The revised manuscript also includes comparison of chemokine protein expression by Olink proteomics, which reveals that CCL2 protein is expressed at very high levels by A3250 but not by SUM149 (revised Fig. 7, new Suppl. Table 11).

2. Fig. 1N: inflammatory pathway, JAK/STAT, angiogenesis not enriched in A3250.

We performed new mRNA-Seq analysis to compare the transcriptome of A3250 cells with another IBC cell line, SUM149, and with non-IBC cells MDA-MB-231.

Gene Set Enrichment Analysis (GSEA) by using the Hallmark collection of molecular signature databases (MSigDB) showed that from the pathways implicated in inflammation, hallmark “TNFA signaling via NFkB” was significantly enriched in A3250. Hallmark “IL6 JAK STAT3 signaling” was also significantly enriched in A3250, indicating that the JAK/STAT3 pathway is activated, in agreement with the previous studies implicating JAK/STAT3 pathway in IBC. Hallmark “Angiogenesis” was not enriched in A3250, consistent with our data showing that A3250 tumors were poorly vascularized.

IPA canonical pathway analysis and analysis of upstream regulators showed remarkable enrichment in pathways associated with inflammation in A3250 cells. From nine canonical pathways differentially activated, eight were associated with inflammation and one with the regulation of tumor microenvironment. Activated upstream regulators also showed striking association with inflammation and regulation of innate immunity. TNF was the top activated upstream regulator, and other main categories included interferon regulatory factors, interferons, and interleukins among others. Taken together, these data demonstrate prominent activation of pathways associated with inflammation and innate immunity in A3250 IBC compared to MDA-MB-231 non-IBC cells, whereas the overall differences in inflammatory pathways between A3250 and SUM149 IBC cells was small.

These new data are described in the Results section “Transcriptomic profiling reveals activation of multiple pathways associated with inflammation in A3250 cells” (page 8-10), in new Figure 3, and in new Suppl. Tables 5, 6, 7, and 8.

3. Its triple-negative status should be evaluated by IHC staining or another standard method.

Agree. In response, we performed IHC for ER, PR and HER2 on A3250 tumors. Breast cancer xenografts MCF-7 and BT474 were used as positive controls for ER, PR and HER2 staining, respectively. Immunostaining shows no detectable ER, PR and HER2 protein in A3250 tumors, indicating that A3250 is a triple-negative cell line. These new results are shown in revised Suppl. Fig. 1 E-J. Main text has been revised as follows (page 5):

“A3250 orthotopic tumors also lacked ER, PR or HER2 protein, in contrast to MCF7 or BT474 tumors that stained positive for ER, PR or HER2, respectively”.

10. *Analysis of gene expression profiles of IBC patients was not compared with non-IBC patients. Therefore, it is not clear whether the result is IBC-specific or not, such as macrophage-rich gene expression and high CCL2 expression.*

Agree. There is as yet no publicly available mRNA-Seq study comparing human clinical IBC and non-IBC transcriptomes. Thus, to examine expression of CCL2 and macrophage-rich gene expression in IBC patient samples, we analyzed microarray data sets.

Analysis of two microarray datasets showed that CCL2 median expression levels were ranked significantly higher in IBCs than in non-IBCs (new Fig. 7F, G; Results page 13, 14). Furthermore, CCL2 was expressed at much higher levels in A3250 IBC cells than in TCGA breast cancer samples not annotated as IBC (n=1098) (new Fig. 7E, Results page 13).

To examine whether human IBCs show enrichment in macrophage genes, we applied a 37-gene signature specific for tumor-associated macrophages (TAMs) in breast cancer, recently reported by the Pollard lab (Cassetta et al., *Human Tumor-Associated Macrophage and Monocyte Transcriptional Landscapes Reveal Cancer-Specific Reprogramming, Biomarkers, and Therapeutic Targets*. Cancer Cell. 2019). Hierarchical clustering revealed two groups of breast cancers: enriched in expression of TAM genes or not enriched. The proportion of IBC samples was significantly higher among breast cancers enriched in TAM genes (60% IBCs and 40% non-IBCs), than among breast cancers not enriched in TAM genes (24.5% IBCs and 75.5% non-IBCs) (p-value of the chi-square test of 0.007) These new results are included in revised Fig. 10A and Results page 17).

11. *In this study, only mRNA expression was examined for chemokines. Their expression at the protein level should be assessed by ELISA or other methods.*

Agree. To examine chemokine expression at the protein level, we used the Olink multiplex immunoassay on supernatants of A3250, SUM149 and MDA-MB-231 cultures. CCL2 protein was detected only in supernatants of A3250, and not in SUM149 or MDA-MB-231 (new Fig. 7D, new Suppl. Table 11), in agreement with RNA expression data (revised Fig. 7C, revised Suppl. Table 10). CXCL5, CCL20 and CXCL1 proteins were present at higher levels in supernatants of A3250 and SUM149 cells relative to MDA-MB-231. These new data are described in the Results, on page 13.

Please note that results of the Olink assay for a given protein are relative and can only be compared across different cell lines. While these results correlated relatively well with qPCR or mRNA-Seq, differences could reflect variations in RNA translational efficiency, protein secretion, and/or stability.

12. *The reason why MDA-MB-231 was used as control is unclear. Is MDA-MB-231 neutrophil-enrich?*

MDA-MB-231 was used as a non-IBC control because it is a triple-negative non-IBC, making it possible to investigate molecular and biological differences between an IBC and non-IBC. We observed that MDA-MB-231 tumors demonstrated neutrophil enrichment, which are new findings that contrast strikingly with the absence of neutrophils in A3250 tumors and SUM149 IBC tumors.

13. *How does the tumor begin to look like when their CCL2 were knocked out? Did they lose the IBC features such as erythema or tumor emboli?*

A3250 CCL2 KD tumors exhibited a striking reduction in erythema, and this effect was independent of tumor size. These data are shown in the new Suppl. Figure 6 and described in Results, on page 15.

Histopathological analysis of cytokeratin-stained tumor sections revealed that the overall invasiveness of A3250 tumors was not decreased upon CCL2 KD (new Suppl. Fig. 7A-F, Results page 15). While there was a lower frequency of tumor emboli in lymphatics upon CCL2 KD, this could be due to the reduction in viable tumor mass and increased necrosis associated with CCL2 KD.

8. All experiments should be done with at least one more different IBC cell.

Agree. We performed new mRNA-Seq experiment for A3250 and SUM149, and used publicly available IBC3 mRNA-Seq data to compare transcriptomes by hierarchical clustering. In more detailed bioinformatics analyses, we compared A3250 with SUM149 and non-IBC MDA-MB-231 for differentially expressed genes, performed comparative analyses of IPA canonical pathways, upstream regulators, and GSEA for select pathways of interest (new Fig.3, new Suppl. Tables 5-8). We also compared A3250 chemokine expression profile to SUM149 and IBC3 by mRNA-Seq (revised Suppl. Table 10).

We also performed new qPCR analyses to compare expression of CCL2 and seven additional chemokines expressed by A3250 with their expression by SUM149 and MDA-MB-231 (revised Fig. 7B, C).

Furthermore, we performed new in vivo experiments to compare tumor characteristics of SUM149 with A3250 including macrophage infiltration by immunostaining with F4/80 and Iba1, with A3250 tumors showing much more florid macrophage infiltration than observed with SUM149 or MDA231 (revised Fig. 4, revised Suppl. Fig. 4). We observed no neutrophils in SUM149 tumors by immunostaining with Ly6G, comparable to A3250 (data not shown). Striking skin erythema associated with A3250 tumors was not seen with SUM149 tumors (new Suppl. Fig. 6). We also observed a lack of dermal lymphatic vessel invasion by SUM149 (revised Suppl. Fig. 2), whereas this was florid in A3250 tumors (Fig. 1). These and other phenotype differences are described in the revised manuscript.

Of note, these results correlated well with the lack of detectable CCL2 expression by SUM149 as compared to the high levels expressed by A3250. Moreover, our findings that these aspects of the florid IBC phenotype were ameliorated by CCL2 knockdown in A3250, strongly argue that overexpression of CCL2 contributes importantly to the IBC phenotype.

We greatly appreciate the constructive comments of the reviewers, which have helped us to further strengthen our results and conclusions. We hope that the revised manuscript is now considered acceptable for publication in Nature Communications.

Sincerely,

Mihaela Skobe

Reviewers' Comments:

Reviewer #1:

Remarks to the Author:

The authors have taken reviewer comments seriously and extensively revised their manuscript, adding additional studies as requested and clarifying all issues that were raised. Well done. I have no further comments.

Reviewer #2:

Remarks to the Author:

The authors addressed the comments comprehensively. However, the novelty of the study remains to be an issue. The cell line itself is a great tool for studying the IBC, but no novel findings that advance the knowledge of inflammatory breast cancer (IBC). Further, is this truly unique to the IBC? The role of CCL2 in breast cancer has been published.

MDA-IBC3 mRNA-seq data was added to the differential gene expression of Fig. 3A as a second IBC cell line. However, they didn't provide any detailed analysis data between A3250 and MDA-IBC3. No data were shown from this comparison.

Fig. 8: shCCL2, only one stable knockdown clone was used both in vitro and in vivo.

Fig. 9: the immune cell profiling of shCCL2 tumor tissues. CCL2 knockdown reduced macrophage infiltration is already well-known—nothing new about it.

Fig 10 doesn't add any points to the manuscript. I don't see the connection with A3250. It seems to stack some data to the manuscript.

POINT-BY-POINT RESPONSE TO THE REVIEWERS

We thank the reviewers for their positive evaluation. Our responses to the reviewers' comments are indicated below:

Reviewer#1: *The authors have taken reviewer comments seriously and extensively revised their manuscript, adding additional studies as requested and clarifying all issues that were raised. Well done. I have no further comments.*

We greatly appreciate Reviewer#1's very positive comments.

Reviewer#2: *The authors addressed the comments comprehensively. However, the novelty of the study remains to be an issue. The cell line itself is a great tool for studying the IBC, but no novel findings that advance the knowledge of inflammatory breast cancer (IBC). Further, is this truly unique to the IBC? The role of CCL2 in breast cancer has been published.*

We appreciate the reviewer#2's recognition that the A3250 line is a great tool for studying IBC, but disagree with their conclusion that our study doesn't advance knowledge of IBC. While there are studies documenting a role for CCL2 in breast cancer, there are no studies documenting that high level of endogenous CCL2 secretion by the tumor cells drives important and unique aspects of the aggressive IBC phenotype. Moreover, we show that high levels of CCL2 and high densities of macrophages are more frequently seen in human IBC than in non-IBC.

Specific comments:

1. MDA-IBC3 mRNA-seq data was added to the differential gene expression of Fig. 3A as a second IBC cell line. However, they didn't provide any detailed analysis data between A3250 and MDA-IBC3. No data were shown from this comparison.

Agree and in response, we have performed detailed analysis to compare A3250 and MDA-IBC3 transcriptomes. We have now included analysis of differentially expressed genes (DEGs), IPA canonical pathways, upstream regulators, and GSEA for select Hallmark pathways of interest (revised Fig. 3, new Suppl. Data 3 and 6).

2. Fig. 8: shCCL2, only one stable knockdown clone was used both in vitro and in vivo.

Agree and in response, we have included new data analyzing effects of an independent shCCL2 on the IBC phenotype of A3250 in vitro and in vivo. In agreement with our data with the shCCL2 #1, new CCL2 KD did not alter tumor cell proliferation and ability to form colonies in vitro, but led to a reduction in viable tumor mass and macrophage densities in vivo (Suppl. Fig. 8). These results further substantiate the results and conclusions of the initial manuscript.

3. Fig. 9: the immune cell profiling of shCCL2 tumor tissues. CCL2 knockdown reduced macrophage infiltration is already well-known, nothing new about it.

There are several novel findings presented in our manuscript. (1) We demonstrate that very high levels of CCL2 expressed by tumor cells are more frequently seen in IBC than in non-IBC; (2) We demonstrate that IBC cells are uniquely dependent on macrophages for tumor growth; (3) We demonstrate that CCL2 importantly contributes to skin erythema, which is characteristic of IBC; and to (4) systemic inflammation.

Our findings of much reduced macrophage infiltration of tumors with CCL2 knockdown correlated with increased necrosis and reduced proliferation as well as amelioration of other markers of the aggressive IBC phenotype. CCL2 production at very high levels was not observed in other breast cancer cell lines analyzed and correlated specifically in the case of A3250 cells with the aggressive IBC phenotype.

4. Fig 10 doesn't add any points to the manuscript. I don't see the connection with A3250. It seems to stack some data to the manuscript.

Fig. 10 adds important data by extending our findings in the A3250 cell model to human IBC clinical samples. We show that IBCs exhibit increased expression of CCL2 by gene expression analysis in comparison to non-IBCs, and that human IBC tissues have high densities of macrophages in juxtaposition to tumor cells. This data are important because they validate our findings from the mouse model.